# Atomically synergistic Zn-Cr catalyst for iso-stoichiometric co-conversion of ethane and CO$_2$ to ethylene and CO

Ji Yang [1,2,12], Lu Wang[3,12], Jiawei Wan [4,12], Farid El Gabaly [5,12], Andre L. Fernandes Cauduro[5], Bernice E. Mills[5], Jeng-Lung Chen [6], Liang-Ching Hsu [7], Daewon Lee[4], Xiao Zhao [4], Haimei Zheng [4], Miquel Salmeron [4], Caiqi Wang [8], Zhun Dong[8], Hongfei Lin [8], Gabor A. Somorjai [9], Fabian Rosner[10], Hanna Breunig [10], David Prendergast [2], De-en Jiang [3,11] ✉, Seema Singh[5] ✉ & Ji Su [1,2] ✉

Developing atomically synergistic bifunctional catalysts relies on the creation of colocalized active atoms to facilitate distinct elementary steps in catalytic cycles. Herein, we show that the atomically-synergistic binuclear-site catalyst (ABC) consisting of Zn$^{\delta+}$-O-Cr$^{6+}$ on zeolite SSZ-13 displays unique catalytic properties for iso-stoichiometric co-conversion of ethane and CO$_2$. Ethylene selectivity and utilization of converted CO$_2$ can reach 100 % and 99.0% under 500 °C at ethane conversion of 9.6%, respectively. In-situ/ex-situ spectroscopic studies and DFT calculations reveal atomic synergies between acidic Zn and redox Cr sites. Zn$^{\delta+}$ ($0 < \delta < 2$) sites facilitate β-C-H bond cleavage in ethane and the formation of Zn-H$^{\delta-}$ hydride, thereby the enhanced basicity promotes CO$_2$ adsorption/activation and prevents ethane C-C bond scission. The redox Cr site accelerates CO$_2$ dissociation by replenishing lattice oxygen and facilitates H$_2$O formation/desorption. This study presents the advantages of the ABC concept, paving the way for the rational design of novel advanced catalysts.

The synergistic effects generated by mixed or supported bifunctional catalysts are often claimed to promote the catalytic performance of traditional heterogeneous catalysts[1–9]. Recently, the construction of synergistic pair-sites with colocalized metal atoms to facilitate distinct elementary steps in the catalytic reaction has been established as a crucial step toward atomically synergistic bifunctional catalyst development[10–12]. The interaction between these adjacent metal atoms,

similar to the metal-support interaction of traditional heterogeneous catalysts[3,13], offers the possibility to modulate their respective electronic structure to further enhance their catalytic activity[11,14–17]. However, precisely controlling their colocalization requires a complicated synthesis process with multiple synthesis steps to load single atoms[10,18]. Moreover, the stability of the well-defined synergistic sites is debatable, especially under harsh reaction conditions such as high

---

[1]Energy Storage and Distributed Resources Division, Lawrence Berkeley National Laboratory, Berkeley, CA, USA. [2]The Molecular Foundry, Lawrence Berkeley National Laboratory, Berkeley, CA, USA. [3]Department of Chemistry, University of California, Riverside, CA, USA. [4]Materials Sciences Division, Lawrence Berkeley National Laboratory, Berkeley, CA, USA. [5]Sandia National Laboratories, Livermore, CA, US. [6]National Synchrotron Radiation Research Center, Science-Based Industrial Park, Hsinchu, Taiwan. [7]Department of Soil and Environmental Sciences, National Chung Hsing University, Taichung, Taiwan. [8]Gene and Linda Voiland School of Chemical Engineering and Bioengineering, Washington State University, Pullman, WA, USA. [9]Department of Chemistry, University of California, Berkeley, CA, USA. [10]Energy Analysis and Environmental Impacts Division, Lawrence Berkeley National Laboratory, Berkeley, CA, USA. [11]Department of Chemical and Biomolecular Engineering, Vanderbilt University, Nashville, TN, USA. [12]These authors contributed equally: Ji Yang, Lu Wang, Jiawei Wan, Farid El Gabaly. ✉e-mail: de-en.jiang@vanderbilt.edu; seema.rose.singh@gmail.com; jisu@lbl.gov

reaction temperatures[18,19]. The benefits and remaining challenges of atomically synergistic bifunctional catalysts motivate us to further explore new synthesis routes, new reaction applications, and their synergistic effects with the goals to accelerate the development of next-generation atomically synergistic catalysts.

Growing $CO_2$ emissions and abundant shale gas reserves have prompted a significant amount of research to explore efficient approaches for co-utilizing the products to produce value-added chemicals[20–22]. Ethane, the second-largest component of shale gas, is an ideal alternative hydrogen source for $CO_2$ conversion. The co-conversion of ethane and $CO_2$ ($C_2H_6 + CO_2 \rightarrow C_2H_4 + CO + H_2O$) is a viable alternative to the ethane steam cracking for ethylene production under the goal of net negative $CO_2$ emissions[21]. Furthermore, iso-stoichiometric co-conversion of ethane and $CO_2$ (ICEC) to ethylene and CO is crucial for direct downstream processes such as the hydroformylation reaction to produce aldehyde[23] and polymerization process to produce polyketones[24–27]. However, the ICEC process lacks a viable catalyst for achieving high ethylene selectivity and $CO_2$ utilization simultaneously. Metal-based catalysts suffer from the inevitable cleavage of C-C bonds through dry reforming pathways and lower the ethylene selectivity[22,28–30]. Oxide-based catalysts exhibit the great merit of preferential C−H bond scission over C−C bond scission pathways but require excessive $CO_2$ cofeeding to reduce ethane adsorption and scavenge the surface H species[31,32]. Specifically, Zn and Cr oxide-based catalysts were widely studied for co-conversion of ethane and $CO_2$. The acidic $Zn^{2+}$ site displayed high activity for C-H bond cleavage in ethane and $CO_2$ activation, requiring the participation of adjacent active sites to form binuclear sites[33–35]. The challenge is that acidified $Zn^{2+}$-H hydride displays a capacity for C-C bond scission of ethane, leading to undesired production of methane. Redox $Cr^{6+}$ sites require lattice oxygen as a H acceptor to dissociate C-H bonds but trigger the formation of less active $Cr^{3+}$ species. The reoxidation of $Cr^{3+}$ to $Cr^{6+}$ is limited by slow O abstraction from $CO_2$[20]. The above analyses thus indicate the need for the development of an atomically-synergistic binuclear Zn−O−Cr site catalyst for ICEC, with Zn facilitating $CO_2$ adsorption and activation to provide O species for $Cr^{6+}$ regeneration and adjacent Cr as an electron donor reducing the acidity of $Zn^{2+}$ to facilitate its activity and selectivity for C-H bond scission[6,36–39]. More generally, exploring a cooperative redox and acid-base catalytic mechanism for ICEC is highly desirable.

In this work, we show the successful fabrication and demonstration of a Zn-O-Cr atomically-synergistic binuclear-site catalyst (ABC) that is highly efficient for ICEC with high ethylene selectivity and utilization of converted $CO_2$ ($U_{CO_2}$). As we will show, compared with pure Zn and Cr catalysts, Zn-O-Cr ABC displays ~1.5 and ~4-fold higher catalytic activity with 100% ethylene selectivity and 99.0% $U_{CO_2}$ under optimized reaction conditions. Key to this high performance is the discovery that Cr facilitates the formation of a $Zn^{\delta+}$ ($0 < \delta < 2$) site to enhance the β-C-H bond cleavage of ethane, while the resulting Lewis base Zn-H$^{\delta-}$ hydride favors $CO_2$ adsorption and activation, and prevents C-C bond scission of ethane. The redox Cr site accelerates $CO_2$ dissociation and facilitates $H_2O$ formation/desorption. The apparent activation energies of ethane conversion and $CO_2$ conversion are ~70.9 and ~74.0 kJ/mol, which demonstrates the rate matching achieved in ICEC.

## Results

### Controlling Zn and Cr coordination structure

Catalysts were synthesized by mixing a constant total amount of zinc (II) acetate and chromium (III) acetate hydroxide on a SSZ-13 zeolite support with varying Zn/Cr molar ratios, followed by direct decomposition at 550°C and then $Na^+$ neutralization of acidic site on supports (Fig. 1a). The catalyst synthesis method developed here will be referred to as the dry-deposition method (The experimental details are summarized in Methods). The catalysts are denoted as $Zn_xCr_y$/SSZ-13, where x/y refers to the ratio of Zn/Cr (x = 1, y = 0; x/y = 1/2, 1, 2, 3, 4;

x = 0, y = 1). The textural properties, composition analysis, and surface acidity of SSZ-13 and synthesized catalysts are characterized by $N_2$ physisorption, X-ray fluorescence (XRF), and attenuated total reflection Fourier transform infrared (ATR-FTIR) measurements (Supplementary Figs. 1-2 and Tables 1-2). Transmission electron microscopy (TEM), scanning TEM (STEM) images, and energy dispersive spectroscopy (EDS) elemental mappings (Fig. 1b and Supplementary Figs. 3-5) demonstrate high dispersion of Zn and/or Cr oxide phases, with no observable sintering of oxide nanoparticles. The X-ray diffraction (XRD) results reveal the absence of spinel $ZnCr_2O_4$ and zincite ZnO for all dry-deposition synthesized catalysts. $Cr_2O_3$ phases with R3c space group were only observed in Cr/SSZ-13 and $Zn_1Cr_2$/SSZ-13 (Supplementary Fig. 6). The control samples with same amount Zn and Cr precursors were prepared via a co-precipitation method (CP), followed by $Na^+$ neutralization. $ZnCrO_x$ nanoparticles are observed in CP-synthesized samples (Supplementary Fig. 7). And a phase transition from spinel $ZnCr_2O_4$ to ZnO (Supplementary Fig. 8) was detected on CP-synthesized catalysts with Zn/Cr ratios varying from 1/2 to 3/1[40–42].

Figure 1c, Supplementary Figs. 9-11 and Supplementary Table 3 show X-ray photoelectron spectroscopy (XPS) results for Zn 2p, Zn LMM Auger and Cr $2p_{3/2}$ spectra. The formation of $Zn^{\delta+}$ ($0 < \delta < 2$) was confirmed through Auger spectra of Zn LMM. The subpeaks at 987.5 and 990.0 eV in Auger spectra of Zn LMM (Supplementary Fig. 10) are assigned to the $Zn^{2+}$ and $Zn^{\delta+}$ ($0 < \delta < 2$), respectively;[43] And the subpeaks at -576 and -580 eV in Cr $2p_{3/2}$ XPS (Supplementary Fig. 11) were assigned to the $Cr^{3+}$ and $Cr^{6+}$, correspondingly[44]. Figure 1c shows the proportion of $Zn^{\delta+}$ $0 < \delta < 2$ and $Cr^{6+}$ relative to the total amount of ($Zn^{\delta+} + Zn^{2+}$) and ($Cr^{6+}+Cr^{3+}$) in $Zn_xCr_y$/SSZ-13 catalysts, respectively. These proportions vary with Zn/Cr ratios, with the highest $Zn^{\delta+}$($0 < \delta < 2$) proportion generated at Zn/Cr ratio of 3/1 and the highest $Cr^{6+}$ proportions produced at Zn/Cr ratios of 3/1 and 4/1. We hypothesize that the proximal electronic interactions between Zn and Cr could modify their respective oxidation states. This is consistent with the Zn $L_3$-edge X-ray absorption near-edge structure (XANES) spectra analyses (Supplementary Fig. 12), which suggests a decrease in the oxidation state of Zn in $Zn_xCr_y$/SSZ-13 samples compared to pure Zn/SSZ-13 sample. The $Zn^{\delta+}$ and $Cr^{6+}$ content in the CP-synthesized $Zn_3Cr_1$/SSZ-13 sample is similar to those of pure Zn/SSZ-13 and Cr-SSZ-13 samples, respectively (Supplementary Fig. 13).

XANES spectra (Fig. 1d and Supplementary Fig. 14) and extended X-ray absorption fine structure (EXAFS) spectra (Fig. 1e, f, Supplementary Figs. 15–18, Supplementary Tables 4-5) were used to study coordination structures of Zn and Cr sites. As shown in Fig. 1d and Supplementary Fig. 14, the electron transition from Zn 1s to Zn 4p unoccupied orbitals (Feature A) is revealed in Zn K-edge XANES spectra; and Cr K-edge XANES spectra exhibits the peak of electron transition from Cr 1s to Cr 3d-O 2p unoccupied orbitals (Feature B). As the Zn/Cr ratio increases from 1/1 to 4/1 in $Zn_xCr_y$/SSZ-13 catalysts, the intensity of feature A decreases, suggesting electron occupation in Zn 4p unoccupied orbitals, while the intensity of feature B increases, which suggests electrons transfer out of Cr 3d-O 2p orbitals near the conduction band minimum. We hypothesize a link between these observations, that electronic charge transfers from Cr 3d-O 2p character in the conduction band to Zn 4p[45,46], due to a strong Zn-Cr interaction. Thus, the charge transfer decreases the oxidative state of $Zn^{2+}$ to $Zn^{\delta+}$ ($0 < \delta < 2$), which is consistent with XPS and Auger results (Fig. 1c). In Fig. 1e, the scattering peaks at 1.50 Å are assigned to Zn-O coordination in the first shell. The pure Zn/SSZ-13 and $Zn_xCr_y$/SSZ-13 samples exhibited a significantly lower intensity for this Zn-O scattering peak than the ZnO reference, suggesting a higher degree of crystal disorder[47,48]. The scattering peaks at -2.85 Å are assigned to Zn-O-Zn coordination in the second shell. Compared to Zn/SSZ-13 and the ZnO reference, $Zn_1Cr_1$/SSZ-13 and $Zn_3Cr_1$/SSZ-13 exhibited no peak for Zn-O-Zn coordination. Instead, a new peak at -3.08 Å should be attributed to the Zn-O-Cr bond[41]. Differently, $Zn_4Cr_1$/SSZ-13 sample has both Zn-O-Cr and Zn-O-Zn bonds. Cr

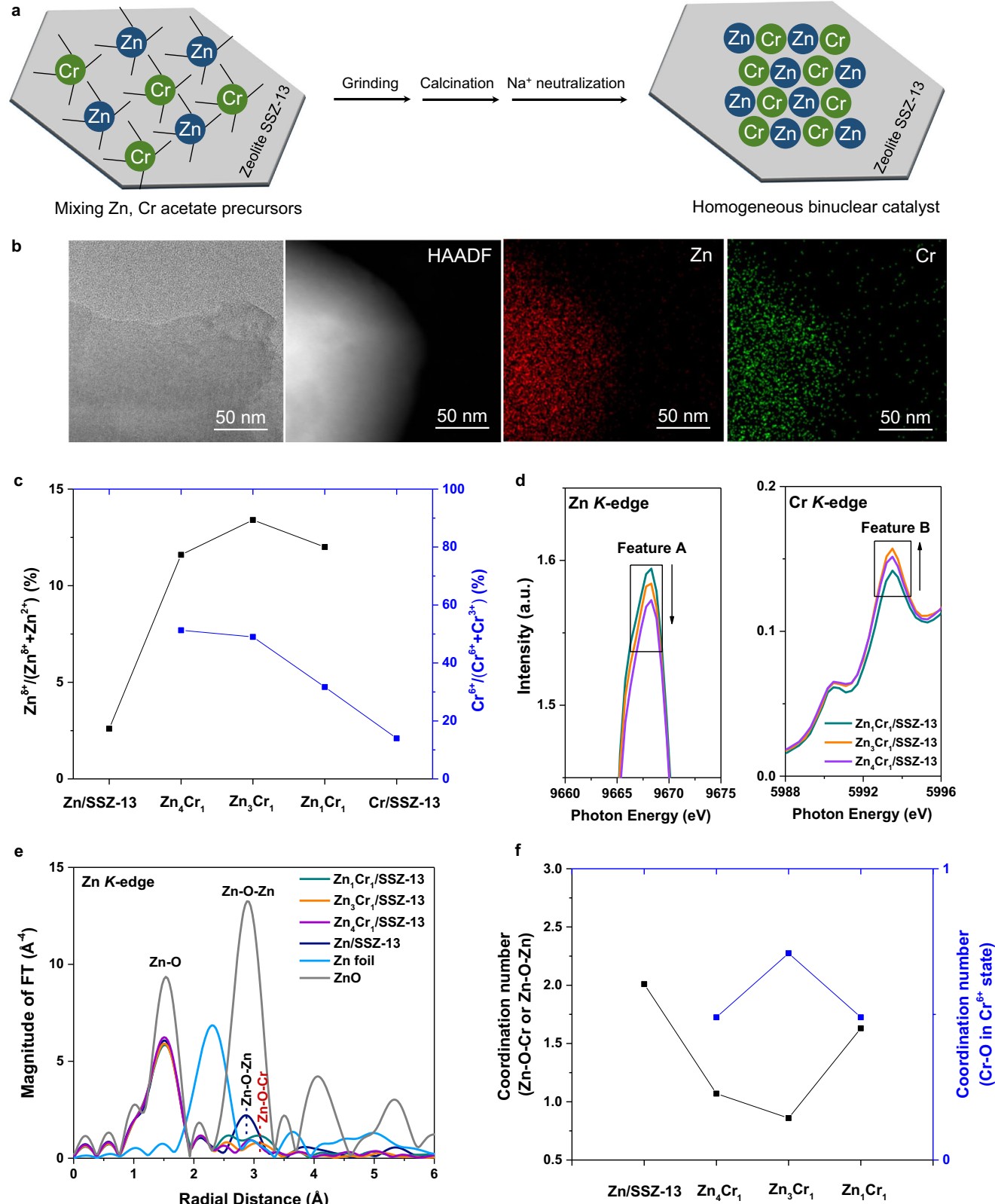

**Fig. 1 | Coordination of Zn-O-Cr sites in atomically-synergistic binuclear-site catalyst (ABC).** **a** Schematic illustration of the fabrication process of Zn-O-Cr ABC. **b** Representative TEM and HAADF-STEM images and EDS mapping of Zn-O-Cr ABC with Zn/Cr ration of 3/1; **c** The proportion of $Zn^{\delta+}$ ($0 < \delta < 2$) (left) and $Cr^{6+}$ (right) with varying Zn/Cr ratios (Results derived from Auger spectra of Zn *LMM* and Cr

$2p_{3/2}$ XPS spectra); **d** Enlarged electron transition features in Zn and Cr *K-edge* XANES spectra; **e** $k^3$-weighted Fourier-transformed extended X-ray absorption fine structure (FT-EXAFS) spectra (Zn *K-edge*) of Zn-O-Cr ABCs with varying Zn/Cr ratios, with Zn foil and ZnO as references; **f** Coordination number (CN) of Zn-O-Cr(Zn) and Cr-O in $Cr^{6+}$ state in Zn-O-Cr ABCs with varying Zn/Cr ratios.

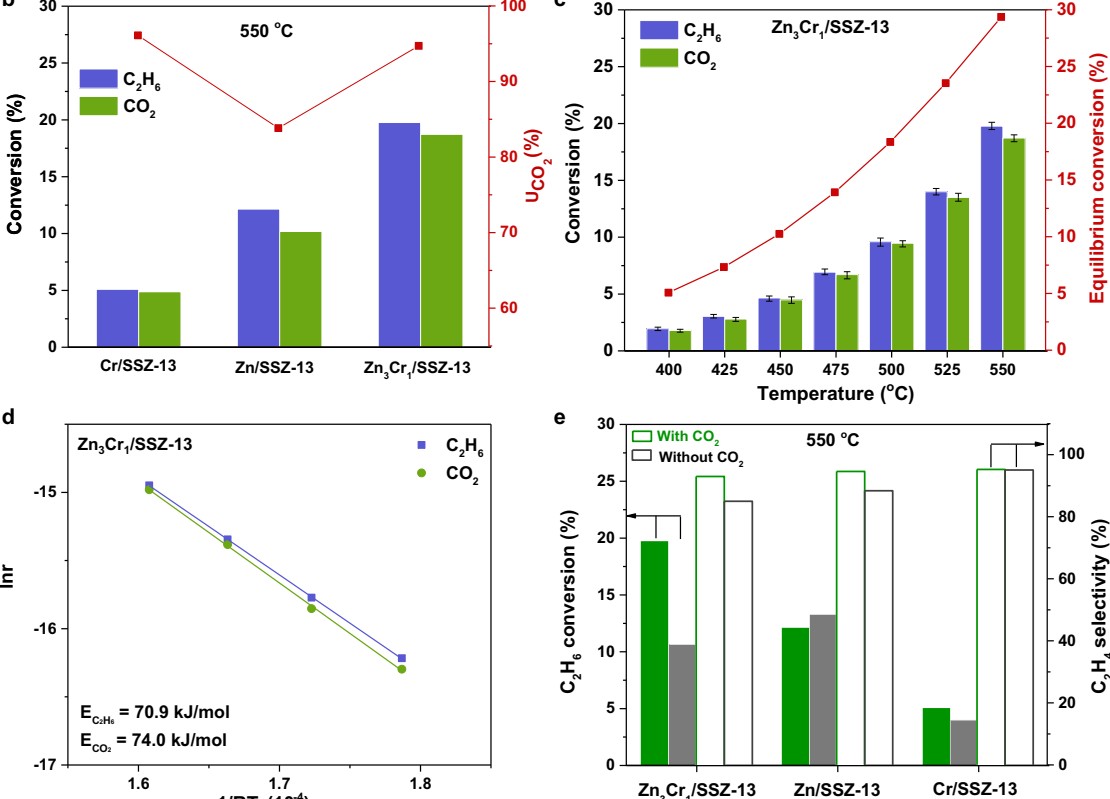

**a**

|  | Cr/SSZ-13 | $Zn_1Cr_2$ | $Zn_1Cr_1$ | $Zn_2Cr_1$ | $Zn_3Cr_1$ | $Zn_4Cr_1$ | Zn/SSZ-13 |
|---|---|---|---|---|---|---|---|
| $C_2H_6$ Conversion (%) | 5.10 | 9.50 | 13.3 | 14.7 | 19.8 | 14.4 | 12.2 |
| $CO_2$ conversion (%) | 4.90 | 9.10 | 12.2 | 13.6 | 18.7 | 11.8 | 10.2 |
| $CO_2$ utilization (%) | 96.0 | 96.0 | 92.0 | 92.5 | 94.4 | 81.5 | 83.6 |
| $C_2H_4$ selectivity (%) | 95.4 | 96.0 | 94.0 | 95.0 | 93.0 | 94.0 | 93.0 |
| TOF of $C_2H_4$ formation (mol $mol_{Zn+Cr}^{-1}$ $h^{-1}$) | 0.20 | 0.38 | 0.52 | 0.58 | 0.77 | 0.57 | 0.48 |

**Fig. 2 | Iso-stoichiometric co-conversion of ethane and $CO_2$ (ICEC). a** Catalytic performance of Zn-O-Cr ABCs (Reaction condition: Temperature = 550 °C; Reactant composition = 5% $C_2H_6$ + 5% $CO_2$ + 90% Ar; WHSV = 7500 mL $g_{cat}^{-1}$ $h^{-1}$); **b** $C_2H_6$ and $CO_2$ conversion, utilization of converted $CO_2$ ($U_{CO_2}$) of Cr/SSZ-13, Zn/SSZ-1$_3$, and $Zn_3Cr_1$/SSZ-13 catalysts in ICEC at 550 °C; **c** Reaction temperature dependent performance of $Zn_3Cr_1$/SSZ-13 catalyst. **d** Arrhenius plots of $Zn_3Cr_1$/SSZ-13 catalyst (obtained when both $C_2H_6$ and $CO_2$ conversions are <10 %). **e** $C_2H_6$ conversion and $C_2H_4$ selectivity over $Zn_3Cr_1$/SSZ-13, Zn/SSZ-13, and Cr/SSZ-13 with (green columns) /without (gray columns) $CO_2$ at 550 °C.

*K*-edge Fourier transformed EXAFS (FT-EXAFS) spectra of Cr-containing samples (Supplementary Fig. 15) displayed Cr-O scattering peaks located at 1.51 Å, which is similar to standard peak of Cr-O scattering 1.50 Å in $Cr^{III}_2O_3$; but as Zn/Cr ratio increase to 3/1 and 4/1, a visible shoulder peak corresponding to standard peak of Cr-O scattering at 1.20 Å in $K_2Cr^{VI}_2O_7$ appeared, indicating that $Zn_3Cr_1$/SSZ-13 and $Zn_4Cr_1$/SSZ-13 samples have two types of Cr-O coordination. Supplementary Tables 4-5 show the EXAFS-derived fitting parameters. $Zn_xCr_y$/SSZ-13 catalyst (*x*/*y* = 1, 3, 4) samples show Zn-O·Cr coordination with a bond distance ~3.4 Å in phase-corrected space, which corresponds to the emergent peak at ~3.08 Å in Fig. 1e[41]. These results provide solid evidence for the formation of a $Zn^{\delta+}$-O-$Cr^{6+}$ structure in $Zn_xCr_y$/SSZ-13. Notice that $Zn_xCr_y$/SSZ-13 samples also exhibited lower coordination numbers (CNs) for Zn-O and Zn-O-Cr compared to the pure Zn/SSZ-

13 sample (Fig. 1f and Supplementary Table 4), which results from Cr colocalization. Especially, for the $Zn_3Cr_1$/SSZ-13 and $Zn_4Cr_1$/SSZ-13 samples, the CNs of Zn-O and Zn-O-Cr are 3.94 and 0.86, and 3.77 and 1.07, respectively. And $Zn_3Cr_1$/SSZ-13 has the highest CN for the Cr-O bond in hexavalent ($Cr^{6+}$) states. Therefore, these results confirm the formation of hetero binuclear $Zn^{\delta+}$-O-$Cr^{6+}$ sites, and $Zn_3Cr_1$/SSZ-13 exhibits the highest population of the $Zn^{\delta+}$-O-$Cr^{6+}$ site.

## Catalytic performance of Zn-O-Cr ABC for ICEC

Figure 2a, b and Supplementary Fig. 19 and Table 6 show the ICEC catalytic performance of Zn-O-Cr ABCs under the reaction conditions of weight hourly space velocity (WHSV) of 7500 mL $g_{cat}^{-1}$ $h^{-1}$ and a $CO_2$/ethane ratio of 1 at 550 °C. The $Zn_3Cr_1$/SSZ-13 ABC catalyst displayed the best performance among all tested catalysts, with the highest $C_2H_6$

conversion (19.8%), $CO_2$ conversion (18.7%), turnover frequency (TOF) of $C_2H_4$ formation (0.77 mol $mol_{Zn+Cr}^{-1}$ $h^{-1}$), as well as excellent $C_2H_4$ selectivity (93.0%) and $U_{CO_2}$ (94.4%). Previous studies have mainly focused on achieving high ethane conversion and ethylene selectivity in co-conversion of ethane and $CO_2$ by co-feeding excess $CO_2$ at $CO_2$/ ethane ratios of 2–6[49–53]. In the ICEC process the $C_2H_4$ selectivity and $U_{CO_2}$ are two important factors for evaluating the success of catalyst development. Utilization of converted $CO_2$ ($U_{CO_2}$) is defined as the ratio of $CO_2$ conversion to ethane conversion on an iso-stoichiometric basis. Compared with previous studies in Supplementary Table 7 and Fig. 21, the $Zn_3Cr_1$/SSZ-13 catalyst displays the highest space-time yield (STY) of $C_2H_4$ formation (0.086 kg $h^{-1}$ $kg_{cat}^{-1}$) and the highest $U_{CO_2}$ at 550 °C, which clearly shows the superiority of Zn−O−Cr ABCs. Furthermore, Fig. 2c and Supplementary Table 7 show $C_2H_4$ selectivity can achieve 100% and $U_{CO_2}$ is 99.0% under the reaction temperature of 500 °C, with ethane conversion of 9.6% and $CO_2$ conversion of 9.5%. These results further demonstrate the advantage of Zn-O-Cr ABCs suitable for ICEC. In ICEC, the reaction system includes the reactants' (ethane and $CO_2$) adsorption, activation, reaction, and products' (ethylene, CO, and $H_2O$) formation, and desorption. The apparent activation energies ($E_a$) for ethane conversion and $CO_2$ conversion are key to evaluating the rate matching of both reactions. Figure 2d shows similar $E_a$ (results were calculated based on Fig. 2c and Supplementary Fig. 22) for ethane dehydrogenation (70.9 kJ/mol) and $CO_2$ hydrogenation (74.0 kJ/mol). This further demonstrates the feasibility and success of the Zn-O-Cr ABCs concept for desired ICEC catalyst development.

Figure 2a and Supplementary Fig. 19 reveal that there is a clear correlation between the proportion of $Zn^{\delta+}$-O-$Cr^{6+}$ sites and ICEC performance. Pure Cr/SSZ-13 catalyst displayed slightly higher ethylene selectivity (95.4%) and higher $U_{CO_2}$ (96.0%) than $Zn_3Cr_1$/SSZ-13 sample, but has the lowest $C_2H_6$ conversion (5.1%), $CO_2$ conversion (4.9%), TOF of $C_2H_4$ formation (0.20 mol $mol_{Zn+Cr}^{-1}$ $h^{-1}$). Increasing Zn contents in Zn-O-Cr ABCs, with Zn/Cr ratios from 1/2 to 3/1, has little effect on ethylene selectivity and $U_{CO_2}$. Instead, increased Zn contents promote higher conversion of $C_2H_6$ and $CO_2$, and leads to increased TOF of ethylene formation. Further increasing the Zn content to Zn/Cr ratio of 4/1, triggers lower ethane (14.4%) and $CO_2$ (11.7%) conversions, resulting in a much lower $U_{CO_2}$ (81.5 %). Pure Zn/SSZ-13 exhibited further decreased $C_2H_6$ (12.1%) and $CO_2$ (10.2%) conversions, compared with $Zn_4Cr_1$/SSZ-13. Therefore, we hypothesize that the $Zn^{\delta+}$ in the binuclear site of $Zn^{\delta+}$-O-$Cr^{6+}$ is the primary active site for $C_2H_6$ dehydrogenation. Interestingly, the $Zn^{\delta+}$ proportion also displays a correlation with $CO_2$ conversion, which indicates that the $Zn^{\delta+}$ sites are also involved in $CO_2$ adsorption, activation, or reaction. In previous studies, binuclear Zn-O-Zn catalysts have been reported to have a high activity in ethane or propane dehydrogenation, but lack the capacity of efficient $CO_2$ activation, leading to insufficient $CO_2$ utilization[35,54,55]. In our study, the $Zn_3Cr_1$/SSZ-13 ABC with the highest amount of $Zn^{\delta+}$-O-$Cr^{6+}$ sites displayed ~1.5 and ~4-fold higher ethane dehydrogenation and $CO_2$ conversion performance than pure Zn and Cr catalysts, respectively (Fig. 2b), which indicates the $Cr^{6+}$ site of $Zn^{\delta+}$-O-$Cr^{6+}$ is also involved in the activation and reaction of $C_2H_6$ and $CO_2$. From the above analyses, we conclude that the unique performance of Zn-O-Cr ABCs for ICEC relies on the atomic synergies within the $Zn^{\delta+}$-O-$Cr^{6+}$ site.

To study the atomic synergies between $Zn^{\delta+}$-O-$Cr^{6+}$ site in ICEC, we compared the $C_2H_6$ dehydrogenation performance of $Zn_3Cr_1$/SSZ-13 with pure Zn/SSZ-13 and Cr/SSZ-13 samples in the presence and absence of $CO_2$. In Fig. 2e, we found the $CO_2$ co-feeding significantly improved $C_2H_6$ conversion (19.8% vs. 10.7%) and $C_2H_4$ selectivity (93.0% vs. 85.0%) for $Zn_3Cr_1$/SSZ-13, with a high $U_{CO_2}$ up to 94.4%. By contrast, for Zn/SSZ−13, $CO_2$ addition fails to enhance its $C_2H_6$ conversion (12.1% vs. 13.3%) but results in an increase in $C_2H_4$ selectivity (94.6% vs. 86.0%). The higher $C_2H_4$ selectivity is due to competitive adsorption of $CO_2$ over $C_2H_6$, preventing the ethane cracking reaction,

which has been reported previously[56]. Compared with Zn/SSZ−13, the improved $C_2H_6$ conversion of the $Zn_3Cr_1$/SSZ-13 catalyst is due to the generation of $Zn^{\delta+}$ in $Zn^{\delta+}$-O-$Cr^{6+}$. For the Cr/SSZ−13 sample, $CO_2$ introduction results in a higher $C_2H_6$ conversion (5.1% vs 4.0%) with a high $U_{CO_2}$ of 96%, but it did not change the $C_2H_4$ selectivity (95.4% vs 95.1%). Cr-based catalysts are reported to catalyze $CO_2$ and ethane conversion through the redox (or MvK) mechanism[20]. $CO_2$ introduction could favor lattice oxygen replenishment to regenerate highly reactive $Cr^{6+}$ species and shift the reaction equilibrium of $C_2H_6$ dehydrogenation, thus leading to a higher activity. These results indicate that a reaction synergy between ethane and $CO_2$ conversions could only occur at the atomically synergistic $Zn^{\delta+}$-O-$Cr^{6+}$ site.

To further validate the superiority of Zn-O-Cr ABCs, we compared the performance of $Zn_3Cr_1$/SSZ−13 prepared by the dry-deposition method with the sample synthesized by the traditional co-precipitation (CP) method. Supplementary Fig. 23 shows that the dry-deposition synthesized $Zn_3Cr_1$/SSZ-13 catalyst exhibited >4-fold higher ethane conversion (19.8% vs 4.5%) and higher $U_{CO_2}$ (94.4% vs 83.2%). The CP-synthesized $Zn_3Cr_1$/SSZ-13 sample was found to contain large Zn/Cr oxide particles and separate $ZnCr_2O_4$ and ZnO phases, resulting in loss of the atomic synergies of the $Zn^{\delta+}$-O-$Cr^{6+}$ site, which we correlate with its poor performance. We also studied the stability and regeneration ability of the $Zn_3Cr_1$/SSZ-13 ABC catalyst. Supplementary Fig. 24 shows that the ethylene selectivity and $U_{CO_2}$ remained nearly 100% during a total of 150 h in 3 cycles. And the decayed conversion of ethane and $CO_2$ can be totally regenerated by oxidative treatment in air at 500 °C. This excellent durability and regeneration ability demonstrates the structural stability of $Zn^{\delta+}$-O-$Cr^{6+}$ site.

## Electronic structure of binuclear $Zn^{\delta+}$-O-$Cr^{6+}$ sites during the reaction

In situ ambient pressure X-ray photoelectron spectroscopy (APXPS) was employed to examine the electronic structure of binuclear $Zn^{\delta+}$-O-$Cr^{6+}$ sites and study the atomic synergies between $Zn^{\delta+}$ and $Cr^{6+}$ sites in ICEC. Figure 3a–c shows ambient pressure spectra indicating C 1s binding energies, Zn *LMM* Auger kinetic energies, and Cr $2p_{3/2}$ binding energies.

Step 1: $Zn_3Cr_1$/SSZ-13 was first tested under 550 °C in an ultra-high vacuum (UHV). The temperature of 550 °C was used to match the reaction temperature in Fig. 2. In Auger spectra of Zn *LMM* (Fig. 3b), both $Zn^{2+}$ (dark blue) and $Zn^{\delta+}$ (green) species were detected with a percentage distribution of 72.4% and 27.6%, respectively. In Fig. 3c, $Cr^{3+}$ (dark red) and $Cr^{6+}$ (light blue) were detected with a proportion of 50.6% and 49.4%, respectively.

Step 2: Then the $Zn_3Cr_1$/SSZ-13 was subjected to simulated reaction conditions of ICEC by co-feeding 50 mTorr $C_2H_6$ and 50 mTorr $CO_2$ under 550 °C. The signals of gaseous $C_2H_6$ and $CO_2$ were detected in Fig. 3a. Notably, adsorbed CO species ($CO_{ads}$) at 285.5 eV were observed[57,58], which indicates the ICEC reaction occurred. Under the reaction conditions, the proportion of $Zn^{\delta+}$ decreased (from 27.6% to 22.8%) and $Cr^{6+}$ ratio increased (from 49.4% to 54.4%). DFT calculations (Fig. 3d) indicate that the oxidation from $Zn^{\delta+}$ to $Zn^{2+}$ is due to the formation of Zn-$H^{\delta}$ hydride during the ethane dehydrogenation, with more negative charge accumulated on $H^{\delta}$. The oxidation of $Cr^{3+}$ to $Cr^{6+}$ may result from lattice oxygen replenishment via $CO_2$ dissociation[20]. These results indicate both $Zn^{\delta+}$ and $Cr^{6+}$ were involved in the ICEC reaction.

Steps 3 and 4 were designed to understand the role of individual $Zn^{\delta+}$ or $Cr^{6+}$ in ICEC. When feeding 100 mTorr $C_2H_6$ (without $CO_2$) in step 3 (Fig. 3a, b), the signals of $CO_2$ and $CO_{ads}$ species disappeared, which indicates that only ethane dehydrogenation could occur. The proportions of $Zn^{2+}$ and $Zn^{\delta+}$ remain similar to the case of co-feeding $C_2H_6$ and $CO_2$ in step 2, indicating the formation of Zn-$H^{\delta}$ hydride during the ethane dehydrogenation. Notice that the proportions of $Zn^{\delta+}$ and $Zn^{2+}$ are comparable between step 1 (UHV) and step 4

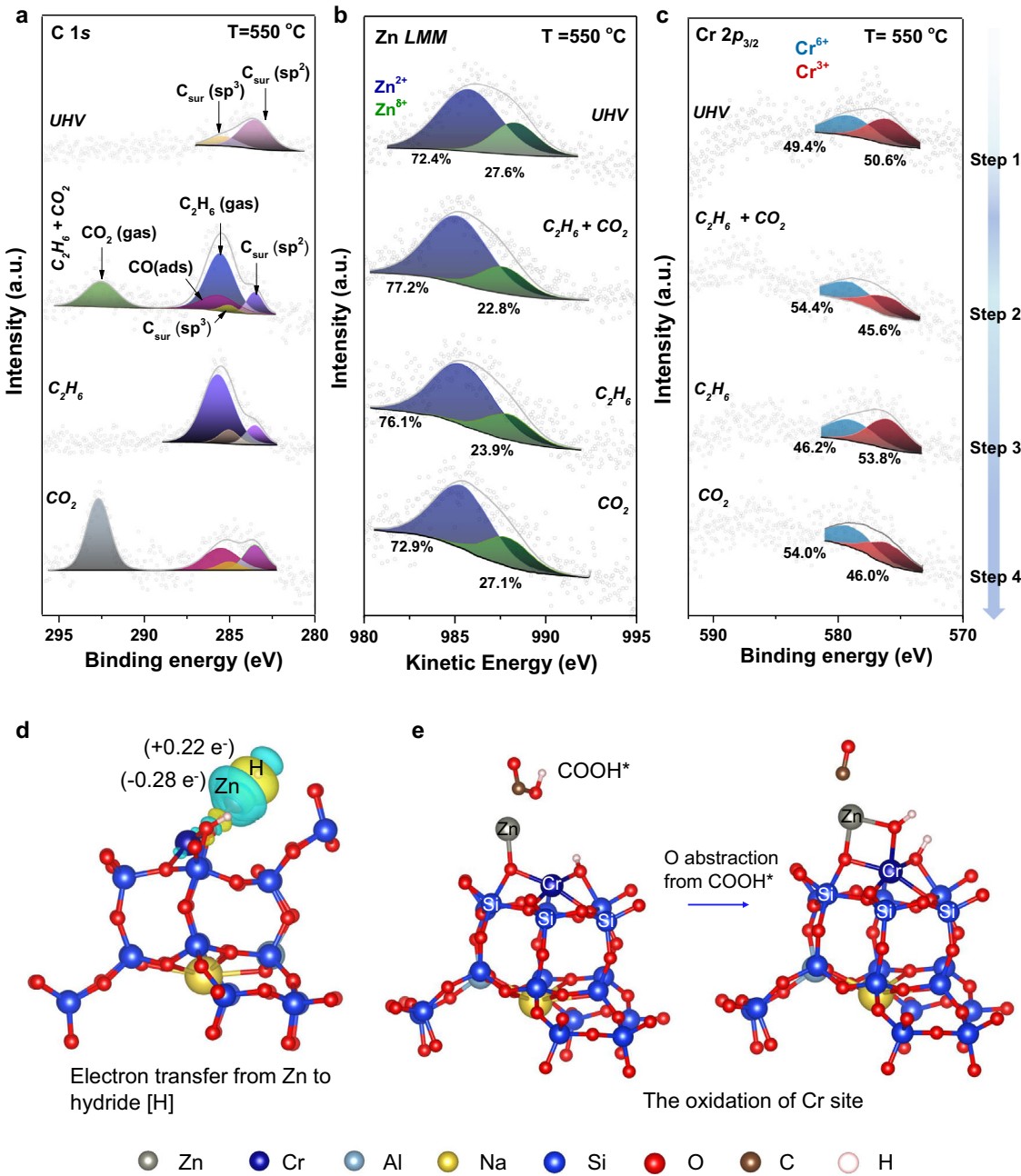

**Fig. 3 | Electronic structure changes of binuclear Zn$^{\delta+}$-O-Cr$^{6+}$ sites.** In situ ambient pressure X-ray photoelectron spectroscopy (APXPS): **a** C 1s spectra, **b** Auger spectra of Zn LMM, and **c** Cr 2$p_{3/2}$ spectra as a function of reaction conditions for Zn$_3$Cr$_1$/SSZ-13. T = 550 °C: ultra-high vacuum (UHV), C$_2$H$_6$ (50 mTorr) + CO$_2$ (50 mTorr), C$_2$H$_6$ (100 mTorr), and CO$_2$ (100 mTorr) in sequence. **d** Electron transfer from Zn to H after the 2$^{nd}$ C-H bond scission in ethane (H: white; Zn: gray), where electron accumulation and depletion are represented by yellow ($\Delta\rho$ = +1 × 10$^{-3}$ e bohr$^{-3}$) and cyan ($\Delta\rho$ = −1 × 10$^{-3}$ e bohr$^{-3}$) respectively. **e** The oxidation of Cr in Zn-O-Cr when decomposing COOH* intermediate.

(feeding 100 mTorr CO$_2$ without ethane), ruling out the likelihood of oxidation of Zn$^{\delta+}$ to Zn$^{2+}$ by CO$_2$ or its derived intermediates.

In the case of Cr 2$p_{3/2}$ XPS spectra (Fig. 3c), the oxidation of Cr$^{3+}$ to Cr$^{6+}$ only occurs when CO$_2$ was fed (steps 2 and 4). Previous studies have demonstrated the oxidation of Cr$^{3+}$ to Cr$^{6+}$ in Cr-based catalysts suffers from sluggish O abstraction from CO$_2$[20,59]. In our cases, the facile reoxidation of Cr$^{3+}$ to Cr$^{6+}$ may be due to the possibility that nearby Zn$^{\delta+}$ facilitate the CO$_2$ activation to enable easier O abstraction. The CO$_{ads}$ species at 285.5 eV were detected in the presence/co-presence of CO$_2$ in the C 1s spectra (Fig. 3a, steps 2 and 4), which further demonstrates facile CO$_2$ dissociation over the Zn$^{\delta+}$-O-Cr$^{6+}$ site. Our DFT calculation (Fig. 3e) indicates that the intermediate of CO$_2$

activation is carboxyl (COOH*). The decomposition of COOH* requires the participation of the Cr site through the formation of a new Cr-O bond (d(Cr-O) = 2.08 Å), which will maintain its high oxidation state. In summary, the APXPS results show that binuclear Zn$^{\delta+}$-O-Cr$^{6+}$ sites serve as atomically synergistic sites for ICEC.

### Atomically synergistic mechanism

We developed atomically synergistic mechanisms for ICEC on binuclear Zn-O-Cr sites (Fig. 4 and Supplementary Fig. 30-35, Supplementary Tables 8-10). Figure 4a shows that the catalytic cycle is initiated by C$_2$H$_6$ adsorption on Zn ([1] [2]), then heterolytic cleavage of the first C-H bond (0.93 eV) by breaking a Zn-O-Cr bond to form Zn-CH$_2$-CH$_3$

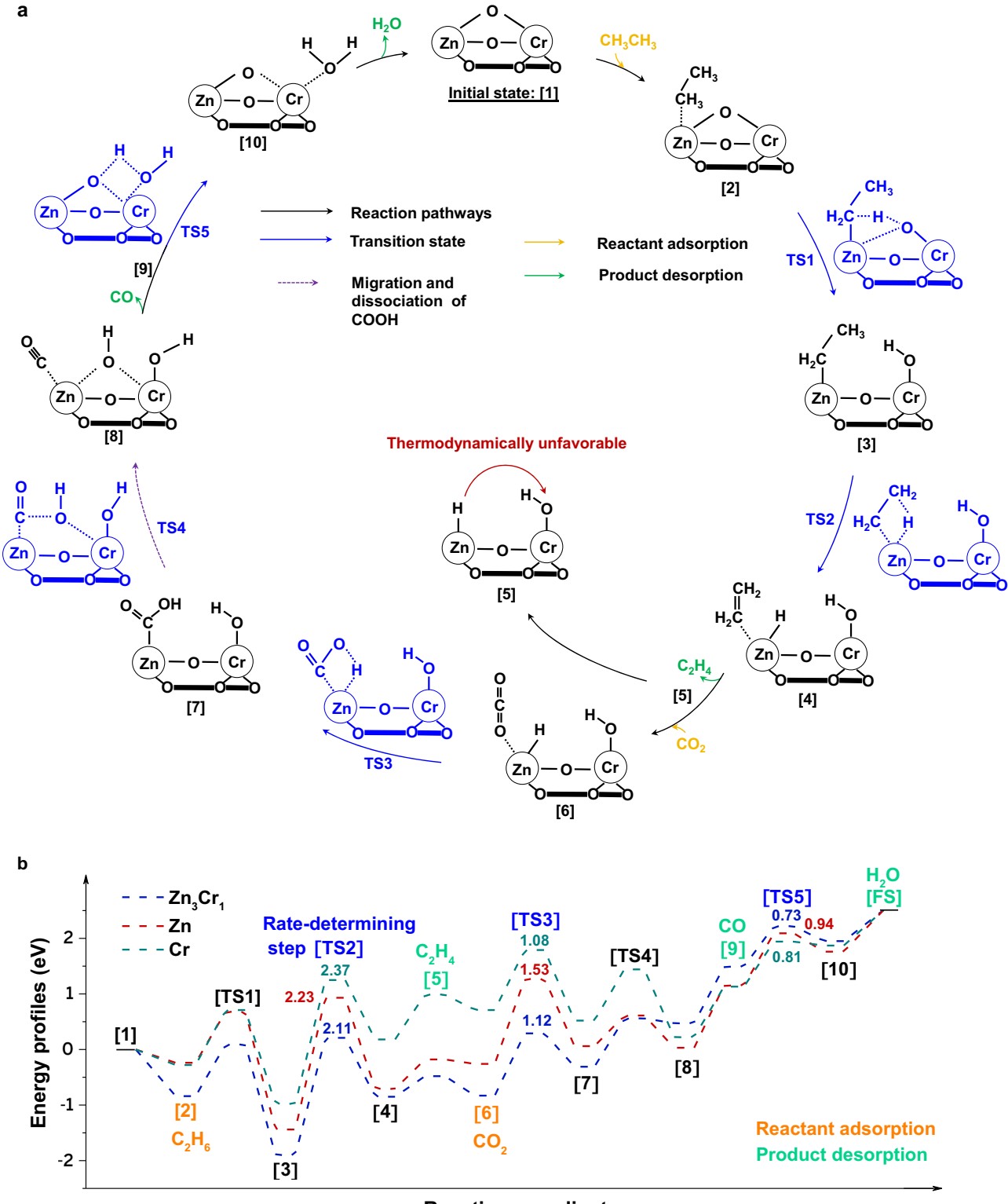

**Fig. 4 | Atomically synergistic mechanism of ICEC over binuclear Zn-O-Cr sites. a** Catalytic cycles of ICEC over binuclear Zn-O-Cr sites of $Zn_3Cr_1$/SSZ–13. **b** The calculated energy profiles of ICEC on $Zn_3Cr_1$/SSZ-13, Zn/SSZ-13, and Cr/SSZ-13.

and Cr-OH ([2] [3]), followed by a homolytic scission of the β-C-H bond (2.11 eV) ([3] [4]), and finally, $C_2H_4$ desorption and Zn-H$^{\delta-}$ hydride formation ([4] [5]). $C_2H_6$ activation at the Cr of Zn-O-Cr sites has higher energy barriers for the first C-H cleavage (1.78 eV vs 0.93 eV) and β-C-H bond dissociation (>3.0 eV vs 2.11 eV), leading to a kinetically unfavorable pathway (Supplementary Figs. 31 and 32). In the second stage

(Fig. 4a), $CO_2$ prefers to adsorb at Zn-H$^{\delta-}$ site (−0.35 eV) than the Cr site (−0.05 eV) through acid-base interaction (Supplementary Fig. 33) ([5] [6])[60,61]; $CO_2$ adsorption energy (−0.35 eV) at Zn-H$^{\delta-}$ site is lower than ethane adsorption (−0.17 eV), which will help prevent the $C_2H_6$ cracking reaction (Supplementary Table 5). Then Zn-H$^{\delta-}$ enables $CO_2$ hydrogenation ([6][7]) to carboxyl (COOH*), followed by CO-OH

cleavage and migration of -OH to be shared with Cr ([7][8]). Finally, after CO desorption from the Zn site and the formation and desorption of $H_2O$ from the Cr site to replenish lattice oxygen ([8][10]), the ICEC catalytic cycle is closed.

The simulated catalytic cycles of Zn/SSZ-13 and Cr/SSZ-13 are shown in Supplementary Figs. 34 and 35, and the calculated energy profiles of ICEC over $Zn_3Cr_1$/SSZ-13, Zn/SSZ-13, and Cr/SSZ-13 are presented in Fig. 4b. The rate-determining step of ICEC is the homolytic cleavage of the β-C-H bond in ethane activation [TS2] (Fig. 4b, Supplementary Table 9). $Zn_3Cr_1$/SSZ-13 exhibited a lower activation barrier (2.11 eV) than Zn/SSZ-13 (2.37 eV) and Cr/SSZ-13 (2.23 eV). Moreover, $Zn_3Cr_1$/SSZ-13 also displayed a much lower energy barrier (1.12 eV, 0.73 eV) for $CO_2$ activation [TS3] and $H_2O$ formation [TS5] than Zn/SSZ-13 (1.53 eV, 0.94 eV). This explained the much higher $CO_2$ conversion and utilization of $Zn_3Cr_1$/SSZ-13 catalyst than Zn/SSZ-13 catalyst (Fig. 2b). We also found that $Zn_3Cr_1$/SSZ-13 exhibited the most favorable energetics of reactants adsorption and products desorption which also helps accelerate catalytic reactions (Supplementary Table 10). Therefore, the kinetically and thermodynamically favorable elementary steps in the ICEC catalytic cycle result in the highest catalytic performance of $Zn_3Cr_1$/SSZ-13. Its superior performance in ICEC is due to the atomic synergies between the acidic Zn site and the redox Cr site of ABC.

## Discussion

In this study, we demonstrated the advantages of an atomically-synergistic binuclear-site catalyst (ABC) synthesized by colocalizing Zn and Cr sites on a zeolite SSZ-13 support (ZnCr/SSZ-13) for the iso-stoichiometric co-conversion of ethane and $CO_2$ (ICEC) reaction. This ZnCr ABC catalyst exhibited exceptional catalytic performance, with 100% ethylene selectivity and 99.0% $U_{CO_2}$ under optimized conditions. The combined results of XAS, AP-XPS, and DFT studies show that the electronic properties and catalytic activity of ZnCr/SSZ-13 can be precisely assigned to unique atomic synergies between neighboring Zn and Cr atoms resulting from colocalization. The redox Cr site facilitates the formation of $Zn^{\delta+}$, which is the active site for easier adsorption and activation of both ethane and $CO_2$. Furthermore, the Cr site accelerates $CO_2$ dissociation, due to its redox properties, and facilitates the formation and desorption of $H_2O$. This combination thus results in rate matching between ethane dehydrogenation and $CO_2$ hydrogenation. Our study highlights the importance of atomic synergy, providing guidance for developing novel catalysts with potential economic and ecological benefits in $CO_2$ conversion and olefin production.

## Methods

Materials synthesis. $Zn_xCr_y$/SSZ-13 (x = 1, y = 0; x/y = 1/2, 1, 2, 3, and 4; x = 0, y = 1) catalysts were prepared by the dry-deposition method[62]. The total amount of Zn and Cr in all catalysts was set at 0.8 mmol per 1 g of support materials. In a typical synthesis, the SSZ-13 zeolite support was pretreated in a vacuum at 120 °C for 3 h to remove moisture. The stoichiometric amount of Zinc (II) acetate and/or chromium (III) acetate hydroxide (Zn + Cr = 0.8 mmol) was thoroughly mixed with 1 g of SSZ-13 zeolites using an analog vortex mixer. The resulting solid mixtures were sufficiently ground in glove box for 20 min. Subsequently, the samples are heated in flowing $N_2$ (100 mL/min) at 550 °C for 3 h at a ramping rate of 2 °C/min and subsequently in flowing air (100 mL/min) for another 3 h. Both $N_2$ and air were purified by moisture trap (Restek). Afterward, the resulting materials were pretreated through $Na^+$-neutralization. Typically, the sample was firstly dispersed in deionized water and then dropwise addition of sodium bicarbonate solution (0.01 M, pH = 8) was performed until the solution reached a pH of 7. The slurry was then collected and dried at 110 °C overnight. The samples were finally calcined in static air at 500 °C for 2 h with a ramping rate of 2 °C/min. The final products were stored in

an $N_2$ box. The control samples of $Zn_xCr$/SSZ-13 (x = 1/2, 1, and 3) were synthesized using the traditional co-precipitation (CP) method according to the reported recipe[41], followed by $Na^+$ neutralization.

Materials characterization. Nitrogen physisorption was performed on the Quantachrome Autosorb iQ2 instrument at 77 K to obtain textural information. The surface area ($S_{BET}$) was determined from the $N_2$ isotherms using the Brunauer-Emmett-Teller (BET) method. X-ray fluorescence (XRF) measurements were performed with a EDAX Orbis Micro-XRF Spectrometer. The attenuated total reflection Fourier transform infrared (ATR-FTIR) measurements were conducted by a Thermo Nicolet iS50 FTIR spectrometer with a diamond crystal ATR module. Powder X-ray diffraction (XRD) analysis was performed on a Rigaku MiniFlex 6 G X-Ray Diffractometer (Cu $K_\alpha$ radiation with wavelength of 1.5406 Å). The transmission electron microscopy (TEM) experiments were performed on a FEI ThemIS aberration-corrected TEM at the National Center for Electron Microscopy (NCEM) of the Molecular Foundry (MF), Lawrence Berkeley National Laboratory (LBNL). The microscope was operated at 300 keV with a Super-X energy dispersive X-ray spectroscopy (EDS) detector, allowing for rapid chemical identification. X-ray photoelectron spectroscopy (XPS) measurement was conducted at a K-Alpha Plus XPS spectrometer (Thermo Scientific), which consists of a monochromatic Al X-ray source (Al $K_\alpha$ = 1,486.68 eV) with variable spot size ranging from 30 microns to 400 microns. Powder samples were placed on a double-sided silver tape and the spectra were acquired using the flood-gun source to account for surface charging. All the spectra were analyzed using the CasaXPS software package.

Zn $L_3$-edge soft X-ray absorption spectroscopy (sXAS) measurement was performed at Beamline 7.3.1 of the Advanced Light Source (ALS) at Lawrence Berkeley National Laboratory (LBNL). sXAS measurement was collected at room temperature through total electron yield (TEY) mode with a probe depth of no more than 10 nm. All the TEY spectra were normalized to the beam flux.

The measurements of Zn K-edge and Cr K-edge XAS spectra including X-ray absorption near edge structure (XANES) and extended X-ray absorption fine structure (EXAFS) were performed at TPS 44 A beamline in National Synchrotron Radiation Research Center (NSRRC) in Taiwan. The data were collected in fluorescence mode by using 7-element silicon drift detector and the Zn and Cr metal foil were used as references for the energy calibration. The data were processed according to standard procedures using Demeter program package.

Ambient-pressure XPS (AP-XPS) analysis was conducted at Sandia National Laboratories (Livermore, CA) using a differentially-pumped Al Kα source (Specs model XR50) with a photon energy of 1486.6 keV. Emitted photo- (and Auger) electrons were detected using a near-ambient pressure hemispherical analyzer (Specs model Phoibos 150) mounted in a custom designed system capable of measuring XPS under sample gas pressures up to 10 Torr. We used baked steel gas lines and leak valves to introduce $C_2H_6$ (99.995% pure) and $CO_2$ (99.999% pure) from Matheson Tri-Gas Inc. XPS/Auger peak locations, widths, and areas were obtained using a Shirley background subtraction and by fitting the data to mixed Gaussian-Lorentzian line shapes using CasaXPS software.

Performance Tests. The catalytic performance of iso-stoichiometric co-conversion of ethane and $CO_2$ (ICEC) was conducted on a continuous fix-bed reactor in the temperature range of 400–550 °C and under ambient pressure, which is held inside an electric furnace with temperature controlled by a K-type thermocouple. In a typical catalytic measurement, a total of 200 mg catalyst diluted by 800 mg sand was loaded into the middle of the reactor plugged by quartz wool on two sides. Before the catalytic test, the catalyst bed was pretreated under a flow of Argon (25 sccm) at 550 °C for 1 h with a ramping rate of 10 °C from room temperature. Afterward, the reactant mixtures consisting of 5% $CO_2$, and 5% $C_2H_6$ balanced with Argon were introduced with a total flow rate of 25 sccm. Argon is used

as an internal standard. The reaction products were analyzed by online GC (Agilent 5890, ShinCarbon ST Packed Columns) equipped with thermal conductivity detector (TCD) and flame ionization detector (FID). To better reveal the reaction kinetics, apparent activation barriers were determined in the temperature range of 400–475 °C (The conversions of $CO_2$ and $C_2H_6$ are less than 10%). The carbon and oxygen balances were within 100 ± 2% for all tests. The conversions of $CO_2$ and $C_2H_6$, $C_2H_4$ selectivity, utilization of converted $CO_2$ ($U_{CO_2}$), $C_2H_4$ yield, the turnover frequency (TOF) of $C_2H_4$ formation, and the space time yield (STY) of $C_2H_4$ formation were calculated as follows (where n denotes molar flow of substance (mol/min), $n_{Zn+Cr}$ means the total molar loading of Zn and/or Cr, $M_{C_2H_4}$ is the molecular weight of $C_2H_4$ (28 g/mol) and $m_{cat}$ stands for catalyst mass (kg)):

$$CO_2 \text{ conversion } (\%) = \frac{n_{CO_2 \text{ input}} - n_{CO_2 \text{ output}}}{n_{CO_2 \text{ input}}} \times 100\% \quad (1)$$

$$C_2H_6 \text{ conversion } (\%) = \frac{n_{C_2H_6 \text{ input}} - n_{C_2H_6 \text{ output}}}{n_{C_2H_6 \text{ input}}} \times 100\% \quad (2)$$

$$C_2H_4 \text{ selectivity } (\%) = \frac{n_{C_2H_4 \text{ output}}}{n_{C_2H_6 \text{ input}} - n_{C_2H_6 \text{ output}}} \times 100\% \quad (3)$$

$$U_{CO_2}(\%) = \frac{CO_2 \text{ conversion}}{C_2H_6 \text{ conversion}} \times 100\% \quad (4)$$

$$C_2H_4 \text{ yield } (\%) = C_2H_6 \text{ conversion} \times C_2H_4 \text{ selectivity} \quad (5)$$

$$TOF = \frac{n_{C_2H_4 \text{ output}} \times 60}{n_{Zn+Cr}} \quad (6)$$

$$STY = \frac{n_{C_2H_4 \text{ output}} \times M_{C_2H_4} \times 60}{1000 \times m_{cat}} \quad (7)$$

It is noted that calculations of $U_{CO_2}$ were also calibrated and examined by the equation:

$$U_{CO_2}(\%) = \frac{n_{CO \text{ output}}}{n_{C_2H_6 \text{ input}} - n_{C_2H_6 \text{ output}}} \times 100\% \quad (8)$$

Computational Methods. All DFT calculations were performed using Vienna ab initio simulation package (VASP)[63,64]. The projector-augmented wave (PAW) method was used to represent the core-valence electron interaction[65,66]. The generalized gradient approximation (GGA) with the Perdew–Burke–Ernzerhof (PBE) exchange-correlation functional was used with D3 dispersion correction[67,68]. The cutoff energy for the planewave basis was 500 eV. To accurately treat the highly localized transition metal 3d orbitals, the spin-polarized DFT + U approach[69,70] was employed: $U_{eff}$ = 4.7 and 3.0 eV were applied to the Zn 3d and Cr 3d state, respectively[71,72]. Electronic energies were converged to within $10^{-4}$ eV and the atomic positions were relaxed until the force on each atom was less than 0.05 eV/Å. The climbing-image nudged elastic band (CI-NEB) method[73] was used to search for the transition states. A hexagonal unit cell of the SSZ-13 molecular sieve[74] was used as the support: $a = b = 13.72$ Å, $c = 14.95$ Å; composition: $Na_1Al_1Si_{35}O_{72}$. Only the Γ-point was used to sample the Brillouin zone. The unit cell contains two hexagonal prisms; two units of $Zn_2O_2$ and $Cr_2O_3$ were placed on the two prisms, to create the models of Zn/SSZ-13 and Cr/SSZ-13, respectively; one $Zn_2O_2$ unit was placed on one of the prisms and one $ZnCrO_2$ unit on the other, to create the model of $Zn_3Cr_1$/SSZ-13 (see Supplementary Fig. 30).

## Data availability
The data that supports the findings of this study are available from the corresponding authors upon request.

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

## Acknowledgements

The research was supported financially by the Division of Chemical Sciences, Geosciences, and Biosciences, Office of Basic Energy Sciences, US Department of Energy (Award No. DE-SC0022273). Spectroscopic and microscopic experiments were performed at the BL7.3.1 at the Advanced Light Source (ALS) and Molecular Foundry at LBNL under contract no. DE-AC02-05CH11231. This research used resources of the National Energy Research Scientific Computing Center, a DOE Office of Science User Facility supported by the Office of Science of the U.S. Department of Energy under contract no. DE-AC02-05CH11231. J.W. and H.Z. acknowledge the support of the U.S. Department of Energy, Office of Science, Office of Basic Energy Sciences (BES), Materials Sciences and Engineering Division under Contract No. DE-AC02-05-CH11231 within the in-situ TEM program (KC22ZH). J.L.C. acknowledges the support of the Ministry of Science and Technology, Taiwan (110-2112-M-213-006). Work at Sandia National Laboratories supported by the U.S. Department of Energy, Office of Science, Basic Energy Sciences (BES) under Field Work Proposal Number 23-024168. Sandia National Laboratories is a multimission laboratory managed and operated by the National Technology & Engineering Solutions of Sandia, LLC, a wholly owned subsidiary of Honeywell International Inc., for the U.S. Department of Energy's National Nuclear Security Administration under contract DE-NA0003525. This paper describes objective technical results and analysis. Any subjective views or opinions that might be expressed in the paper do not necessarily represent the views of the U.S. Department of Energy or the U.S. Government. D.L. acknowledges the Kwanjeong Study Abroad Scholarship from the KEF (Kwanjeong Educational Foundation) (KEF-2019).

## Author contributions

J.S. and S.S. designed the experiments and D.-E.J. designed the DFT calculations. J.Y. synthesized the catalysts and evaluated the catalytic performance, and analyzed the data. L.W. did the DFT calculations. J.W. performed TEM experiments for sample screening, and analyzed the data. F.E.G. and A.F.C. did APXPS experiments and analyzed the data. B.E.M. performed XRF measurements. J.L.C. and L.C.H. performed the XAFS measurements and analyzed the data. D.L. did XRD measurements. X.Z. did XPS experiment. C.W. performed $N_2$ physisorption measurements. Z.D. contributed to characterization analysis. F.R. and H.B. performed thermodynamic limit analysis. H.Z., M.S., G.A.S., H.L. and D.P. provides suggestions with project design. J.S., S.S., D.-E.J., J.Y., L.W., J.W. and F.E.G. wrote the manuscript, and all the authors contributed to the overall scientific interpretation and edited the manuscript.

## Competing interests

The authors declare no competing interests.
