## [Peer Review File · Nature Communications]

REVIEWER COMMENTS

Reviewer #1 (Remarks to the Author):

The authors touched the important topic of ethane and CO₂ co-conversion to added-value products, i.e., ethylene and CO. However, the novelty of their work is not properly highlighted, while there are some important issues to be addressed throughout the manuscript. Some comments/questions that could help the authors improve their manuscript:

Abstract: In my opinion the sentence “CO₂ utilization can reach 100%” is misleading. CO₂ utilization equal to 100% would mean full conversion of CO₂ to added value products, such as CO. However, the table in figure 2 a clearly shows that CO₂ conversion does not exceed 18.7% for Zn₃Cr₁ catalyst. The authors should replace the “CO₂ utilization” term over the manuscript. CO₂ conversion and CO selectivity could be reported.

Abstract: Reporting only the ethylene selectivity has no meaning for the community. The selectivity values should be always accompanied by conversion values (C₂H₄ selectivity at specific C₂H₆ conversion).

Figure 2 a: Can the authors add the thermodynamic equilibrium conversions of CO₂ and C₂H₆? Also, it is not clear the conditions at which the performance reported in Figure 2 a was attained. For example, comparing the column of Zn₃Cr₁ in Figure 2 a and Figure 2 c, the CO₂ and C₂H₆ conversion values are different.

Figure 2c, Figure S.20 and Table S.4 highlight that the XC₂H₆/XCO₂ ratio is slightly > 1. This could imply that C₂H₆ is also participating in another reaction, i.e., deep dehydrogenation and carbon deposition. Did the authors check the carbon deposition of the used catalysts?

Figure S.21: the authors should re-scale the y axis. The deactivation is hardly visible. What is the origin of the observed deactivation?

Supplementary table 4 does not include a complete literature review. Important contributions are missing:

- doi.org/10.1016/j.chempr.2020.07.011

- doi.org/10.1021/acscatal.2c05338
- doi.org/10.1021/ie9007387

Reviewer #2 (Remarks to the Author):

This is a well-written manuscript dealing with an important research topic, the simultaneous upgrading of CO₂ and C₂H₆ to produce CO, C₂H₄ and H₂O. The abundance of C₂H₆ in shale gas reserve makes it an ideal source of H to reduce CO₂; this reaction is potentially advantages over CO₂ reduction using molecular H₂, which generates CO₂ as a byproduct when H₂ is produced from hydrocarbon reforming. Although the reaction of CO₂ + C₂H₆ has been the topic of many recent studies, the current manuscript provides insights into the design of a dual-site Cr-Zn catalyst to achieve nearly 100% C₂H₄ selectivity. The combined experimental and theoretical results also provide useful insights into the reaction mechanism. Before the manuscript can be accepted, the authors should consider the following two suggestions:

1. It is important to compare selectivity at comparable conversions. For example, the results in Table 1 compares the selectivity of Cr/SSZ-13 (5.1% C₂H₆ conversion) and Zn₃Cr1 (19.8% C₂H₆ conversion). Comparing selectivity at conversions that are a factor of 4 different can result in misleading conversions. Because the Arrhenius plot in Figure 1 contains results with less than 10% C₂H₆ conversion, it is likely that the authors already have results at comparable conversions for Cr/SSZ-13 and Zn₃Cr1. The authors should add these results in Table 1 to illustrate that the selectivity is not affected by C₂H₆ conversion.
2. The XANES and EXAFS results provided important information for the authors to support their proposed active sites. However, these measurements appear to be performed under ex-situ conditions. It is possible that the oxidation states of Zn and Cr, as well as the local coordination of Zn, would change under the reaction conditions of CO₂ + C₂H₆. It might be difficult to obtain sufficient beamtime to characterize all the catalysts under in-situ conditions. However, it is important to measure at least one, the best-performing catalyst, under in-situ conditions to confirm that the proposed oxidation states and local coordination environment do not change under reaction conditions.

Reviewer #3 (Remarks to the Author):

The submitted manuscript by Yang et al. describes a very interesting piece of work around true catalyst design for a CO₂ activation reaction, which is still not as prominent in the public domain but holds great potential as it does not require the production of green hydrogen as the RWGS does. The authors

develop the concept of a bi-nuclear Zn-Cr site, provide a synthesis route, extensive characterization and DFT modelling to elucidate the novel catalyst and its workings. To combine all these aspects in a single paper must be applauded. It is a challenge for a single reviewer to engage with the different areas equally and this reviewer will therefore focus primarily on the testing of the catalyst, interpretation of results and description thereof.

Some comments on the manuscript which are believed to further improve the work.

The work encompasses on a high level many crucial aspects of modern catalysis research and design. However a thermodynamic analysis is missing. This could be of special interest as the authors operate the reaction with a dilution of 90% inert. This dilution will have an effect of the thermodynamic limit of the reaction.

It is also challenging for the reader to get a clear picture of the performance of the produced catalyst as the authors continuously switch between experimental performance at 500 and 550C. Stability information is provided to some extent at the lower temperature, but not at 550C. It is also not clear in most cases at what time on stream performance data is collected. It is well reported that CO₂-ODH catalysts commonly suffer from deactivation over the initial 12 to 24 hours, so catalytic performance in the initial hours TOS might not be sufficiently representative.

We would also urge the authors to provide more detail on the product distribution. What is the balance in selectivity if 100% is not reached. I would also prefer if the exact value of the CO₂ utilization is provided and not the the approximately 100%. The authors provide sufficient accuracy in the other data presented.

In Table S4, the authors compare performances of their own design with other reported catalyst systems. One class that has recently attracted some attention, namely molybdenum carbides is only listed in its bulk form. There have been a number of publications recently from different groups suggesting that supported molycarbide does preform much better. It would be beneficial for the manuscript and future readers if this development is reflected.

Reviewer #4 (Remarks to the Author):

The paper focuses on atomically synergistic binuclear-site Zn-Cr zeolitic catalyst for co-conversion of ethane and CO₂.

The study presents advanced material characterization data, modelling, catalytic performances in co-conversion of ethane and CO₂.

The insight in material, the discussions involving modelling and advanced characterization, the selection of the reaction and the reaction conditions to highlight the synergy between Cr-Zn, and the observed phenomena are interesting and warrant publication.

However, there are several comments raised that require attention.

1. The authors report synthetic procedure, XRD and a lot of high-level characterization data of the metal-phases by XAFS, XPS, HRTEM and HAADF-STEM etc. This information is only valuable if the basic characterization of the materials would be given. However, the article is lacking the information about basic characteristics which is typically used by experts in the field, especially for the zeolitic phase. So, the readers of the article will not be able to have any points of comparison of the classical prior art materials with the one, which is reported in this contribution. Without this information, this contribution would not be very useful for the scientific community (difficult to reproduce the data).

One would expect to see values for

- basic porosity characteristics of samples with different metal ratios,
- form of zeolite before modification (H⁺, NH₄⁺, Ca, K etc),
- crystal size of zeolite,
- acidity (any characteristics, i.e. OH-groups by FTIR or NMR).

These values also are fundamental for the explanation of the catalytic data.

2. The concentration of the metals deposited on the samples was not measured (not controlled) by elemental analysis. It was assumed that the total amount of Zn and Cr, which was added to the synthetic mixture as acetates, would be deposited on the zeolites. It is surprising to perform so many advanced measurements for the samples without controlling / knowing the exact chemical composition of the materials.

3. The authors report significantly better catalytic activity, selectivity, stability in co-conversion of ethane and CO₂ vs. the recent publications for the same reaction under similar operating conditions (Appl. Catal. B Environ. 304, 120947 (2022), ACS Catal. 11, 2819–2830).

However, the catalytic data were produced at very low partial pressures of ethane and CO₂. The reported mass balance is 100% ±2 % (based on internal standard Ar). However, the internal standard was used in 16 times excess relative to the ethane and CO₂ in the feed (5/5/80). Under such a condition, the variation of flow of the internal standard of 1% would lead to a significant change in measured catalytic performances. So, under such extreme conditions, a comparison of the data between different labs should be taken with precautions.

Nevertheless, under the reported conditions, the obtained results look not in a conflict with thermodynamics equilibrium (ethylene yield).

One should also mention that the catalytic performance is reported at 500-550°C, which is 100°C-250°C lower than the typical temperature range in the classical literature for the same reaction (Energy & Fuels 2004, 18, 1126-1139, table 2). The reason for that is in the "extreme" catalytic conditions (very low

partial pressure). These conditions may be very beneficial for demonstrating the synergistic effect between Zn and Cr but wasn't used in the classic literature because of the limited practical significance.

So, despite the catalysts show a very good performances, this is a not yet a breakthrough in CO₂ utilization. One should see the reaction as fine characterization of materials properties and as a demonstration of a synergy between Zn-Cr on SSZ-13 zeolite.

So, the material part remains interesting if the additional basic characterization would be provided.

Response to Reviewer 1

The authors touched the important topic of ethane and CO₂ co-conversion to added-value products, i.e., ethylene and CO. However, the novelty of their work is not properly highlighted, while there are some important issues to be addressed throughout the manuscript. Some comments/questions that could help the authors improve their manuscript:

Response: We are grateful for the reviewer's efforts in evaluating the manuscript and providing valuable comments. In this study, we demonstrated a concept of ICEC reaction based on atomically synergistic catalyst design and fabrication. The co-conversion of ethane and CO₂ ($C_2H_6 + CO_2 \rightarrow C_2H_4 + CO + H_2O$) is a viable alternative to the ethane steam cracking for ethylene production under the goal of net negative CO₂ emissions. Ethane, the second-largest component of shale gas, is an ideal alternative hydrogen source for CO₂ conversion. Furthermore, ICEC to ethylene and CO is crucial for direct downstream processes such as the hydroformylation reaction to produce aldehyde and polymerization process to produce polyketones.

Revision:

In the introduction, we added the sentences:

“Ethane, the second-largest component of shale gas, is an ideal alternative hydrogen source for CO₂ conversion”.

1. *Abstract: In my opinion the sentence “CO₂ utilization can reach 100%” is misleading. CO₂ utilization equal to 100% would mean full conversion of CO₂ to added value products, such as CO. However, the table in figure 2a clearly shows that CO₂ conversion does not exceed 18.7% for Zn₃Cr₁ catalyst. The authors should replace the “CO₂ utilization” term over the manuscript. CO₂ conversion and CO selectivity could be reported.*

Response: Thanks for the reviewer's comments and sorry for the confusion. The “CO₂ utilization” was defined as the ratio of CO₂ conversion to C₂H₆ conversion to evaluate the iso-stoichiometric co-conversion of ethane and CO₂. We replaced the “CO₂ utilization” to “Utilization of converted CO₂” (denoted as U_{CO_2}) and updated our manuscript.

2. *Abstract: Reporting only the ethylene selectivity has no meaning for the community. The selectivity values should be always accompanied by conversion values (C₂H₄ selectivity at*

specific C₂H₆ conversion).

Response: Thanks for the reviewer's suggestions. We have added the specific C₂H₆ conversion in the abstract (highlighted in red).

Revision:

“Ethylene selectivity and utilization of converted CO₂ in ICEC can reach 100 % and 99.0% under 500 °C at ethane conversion of 9.6 %, respectively”.

3. *Figure 2a: Can the authors add the thermodynamic equilibrium conversions of CO₂ and C₂H₆? Also, it is not clear the conditions at which the performance reported in Figure 2a was attained. For example, comparing the column of Zn₃Cr₁ in Figure 2a and Figure 2c, the CO₂ and C₂H₆ conversion values are different.*

Response: Thanks for your constructive comments and suggestions. We added the thermodynamic equilibrium conversions of ICEC in Fig. 2c. The results show that C₂H₆ and CO₂ conversions of our study are within thermodynamics equilibriums. And we added the descriptions of the reaction conditions of Fig. 2a to the caption and re-edited the Fig. 2c by including catalytic data of ICEC for Zn₃Cr₁/SSZ-13 under 525 and 550 °C. During the re-editing of Fig. 2, we also re-scaled the y axis of Fig. 2b, 2c, and 2e to make sure that the CO₂ and C₂H₆ conversion values can be easily compared.

For the convenience of review, please check the updated Fig. 2 below.

Revision:

Fig. 2. Iso-stoichiometric co-conversion of ethane and CO₂ (ICEC). **a**, Catalytic performance of Zn-O-Cr ABCs (Reaction condition: Temperature = 550 °C; Reactant composition = 5% C₂H₆ + 5% CO₂ + 90% Ar; WHSV = 7500 mL g_{cat}⁻¹ h⁻¹); **b**, C₂H₆ and CO₂ conversion, utilization of converted CO₂ (U_{CO_2}) of Cr/SSZ-13, Zn/SSZ-13, and Zn₃Cr₁/SSZ-13 catalysts in ICEC at 550 °C; **c**, Reaction temperature dependent performance of Zn₃Cr₁/SSZ-13 catalyst. **d**, Arrhenius plots of Zn₃Cr₁/SSZ-13 catalyst (obtained when both C₂H₆ and CO₂ conversions are <10 %). **e**, C₂H₆ conversion and C₂H₄ selectivity over Zn₃Cr₁/SSZ-13, Zn/SSZ-13, and Cr/SSZ-13 with (green columns) /without (gray columns) CO₂ at 550 °C.

4. *Figure 2c, Figure S.20 and Table S.4 highlight that the $X_{C_2H_6}/X_{CO_2}$ ratio is slightly > 1 . This could imply that C_2H_6 is also participating in another reaction, i.e., deep dehydrogenation and carbon deposition. Did the authors check the carbon deposition of the used catalysts?*

Response: Thanks for your insightful comments and suggestions.

Figure 2c, Supplementary Figure 20 and Table 4 show that the $X_{C_2H_6}/X_{CO_2}$ ratio is slightly > 1 , in other word that the U_{CO_2} (Utilization of converted CO_2) is slightly $< 100\%$, is due to that ICEC reaction on $Zn_3Cr_1/SSZ-13$ is accompanied with a small fraction ($\sim 1\%$ at $500\text{ }^\circ\text{C}$) of direct C_2H_6 dehydrogenation ($C_2H_6 \rightarrow C_2H_4 + H_2$). This was confirmed by our product analysis results.

We also checked the carbon deposition of the used catalysts. As shown in APXPS spectra of C1s below, in ultra-high vacuum (UHV) condition before cofeeding C_2H_6 and CO_2 , very small amount of surface carbon species (sp^2 and sp^3 carbon) were observed on fresh $Zn_3Cr_1/SSZ-13$, which is comparable to that in UHV after pumping out C_2H_6 and CO_2 . This suggests that the accumulation of carbon species on catalyst surface is negligible during ICEC reaction. We updated the APXPS spectra of C1s in Supplementary Information as Supplementary Fig. 28. To further examine possible carbon deposition during reaction, we performed XPS measurement on fresh and spent $Zn_3Cr_1/SSZ-13$ catalysts (after stability test for 150 h). As shown in the Figure below (Supplementary Fig. 27), the intensity of C 1s XPS spectra for spent $Zn_3Cr_1/SSZ-13$ is similar to that of fresh catalyst. This indicates that carbon deposition on $Zn_3Cr_1/SSZ-13$ is negligible during ICEC reaction. This conclusion is consistent with the color change between fresh and spent catalyst (after stability test for 150 h) in Supplementary Fig. 27.

In all, a slightly > 1 of $X_{C_2H_6}/X_{CO_2}$ ratio (Or $U_{CO_2} < 100\%$) is because a small fraction of direct C_2H_6 dehydrogenation ($C_2H_6 \rightarrow C_2H_4 + H_2$) occurred in addition to ICEC reaction on $Zn_3Cr_1/SSZ-13$. We added obtained XPS results as Supplementary Fig. 27 and new analysis in the Supplementary Information.

Revision:

Supplementary Fig. 28. In situ ambient pressure X-ray photoelectron spectroscopy (APXPS): C 1s spectra in UHV before cofeeding C_2H_6 and CO_2 and in UHV after pumping out C_2H_6 and CO_2 .

Supplementary Information, Supplementary Fig. 28, we added the sentence “As shown in Supplementary Fig. 28, in ultra-high vacuum (UHV) condition before cofeeding C_2H_6 and CO_2 , very small amount of surface carbon species (sp^2 and sp^3 carbon) were observed on fresh $Zn_3Cr_1/SSZ-13$, which is comparable to that in UHV after pumping out C_2H_6 and CO_2 . This suggests that the accumulation of carbon species on catalyst surface is negligible during ICEC reaction”.

Supplementary Fig. 27. C 1s XPS spectra and photos for fresh and spent $Zn_3Cr_1/SSZ-13$ catalyst.

Supplementary Information, Supplementary Fig. 27, we added the sentence “As shown in Supplementary Fig. 27, the intensity of C 1s XPS spectra for spent $\text{Zn}_3\text{Cr}_1/\text{SSZ-13}$ is similar to that of fresh catalyst. This indicates that carbon deposition on $\text{Zn}_3\text{Cr}_1/\text{SSZ-13}$ is negligible during ICEC reaction at working conditions. A change of color from an initial light gray to light green yellow (chartreuse) for catalyst powder after stability tests also excludes the carbon deposition during ICEC reaction”.

5. *Figure S.21: the authors should re-scale the y axis. The deactivation is hardly visible. What is the origin of the observed deactivation?*

Response: Thanks for your comments and suggestions. We re-scaled the y axis in Supplementary Fig. 21 (which is now updated as Supplementary Fig. 24). As shown in updated version, the observed deactivation occurs in the reaction induction stage (< 15 hours). After that, CO_2 and ethane conversions remain almost stable (the conversions slightly decreased from 7.5% in 15h to 7.0% in 50h). To further study the origin of observed catalyst deactivation, we performed TEM and XPS measurement on spent catalysts (after stability test for 150h). We updated TEM images and XPS results as Supplementary Figs. 25 and 26 in the Supplementary Information. As shown in TEM image in the Supplementary Fig. 25, the sintering of ZnO and CrOx was observed after stability tests. XPS results in Supplementary Fig. 26 show that the relative ratios of Zn^{2+} is increased and the relative ratio of Cr^{6+} is decreased for spent catalyst compared to fresh catalyst (Zn^{2+} : 90.6% vs. 86.6%; Cr^{6+} : 36.7% vs. 49.0%). Therefore, we think the part of loss of the atomic synergies of the $\text{Zn}^{\delta+}\text{-O-Cr}^{6+}$ site results in the slight deactivation of $\text{Zn}_3\text{Cr}_1/\text{SSZ-13}$ catalyst during ICEC reaction.

Revision:

Supplementary Fig. 24. On-stream curves of C_2H_6 , CO_2 conversion & the space time yield (STY) of C_2H_4 formation over $Zn_3Cr_1/SSZ-13$ ABC catalyst.

Supplementary Information, Supplementary Fig. 24, we added the sentence “The observed deactivation in the initial induction period should result from the part of loss of the atomic synergies of the $Zn^{\delta+}-O-Cr^{6+}$ site (See Supplementary Figs. 25-28)”.

Supplementary Fig. 25. Representative TEM images of spent $Zn_3Cr_1/SSZ-13$ catalyst after

stability test.

Supplementary Information, Supplementary Fig. 25, we added the sentence “As shown in TEM image in the Supplementary Fig. 25, the sintering of ZnO and CrOx was observed after stability tests.

Supplementary Fig. 26. Cr 2p XPS and Zn Auger spectra for fresh and spent Zn₃Cr₁/SSZ-13 catalyst after stability test.

Supplementary Information, Supplementary Fig. 26, we added the sentence “XPS results in Supplementary Fig. 26 show that the relative ratios of Zn²⁺ is increased and the relative ratio of Cr⁶⁺ is decreased for spent catalyst compared to fresh catalyst (Zn²⁺: 90.6% vs. 86.6%; Cr⁶⁺: 36.7% vs. 49.0%). Therefore, we think the part of loss of the atomic synergies of the Zn^{δ+}-O-Cr⁶⁺ site results in the slight deactivation of Zn₃Cr₁/SSZ-13 catalyst during ICEC reaction”.

6. *Supplementary table 4 does not include a complete literature review. Important contributions are missing:*

- doi.org/10.1016/j.chempr.2020.07.011
- doi.org/10.1021/acscatal.2c05338
- doi.org/10.1021/ie9007387

Response: Thanks for your comments and suggestions.

The co-conversion of CO₂ and ethane encompasses a wide and dynamic research area. In Supplementary Table 4 (which has been updated to Supplementary Table 7), we summarized the prior research efforts involving non-noble metal catalysts, specifically under the reaction condition

of a CO₂/C₂H₆ ratio of 1 to compare with our ICEC process. As shown in Supplementary Table 7, we added the relevant results from ref. 11, 16, 17, 18 and 19. Compared with the previous results, our non-noble-metal and atomically synergistic catalyst displayed advantages in achieving higher ethylene selectivity and U_{CO₂}.

Revision:

Supplementary Table 7. Summarization of catalytic behaviors over previously reported catalysts at CO₂/C₂H₆ ratio of 1.

Catalysts	Loading (wt%)	T (°C)	Reactants compositions (C ₂ H ₆ :CO ₂ :Ar/He)	WHSV (mL g _{cat} ⁻¹ h ⁻¹)	C _(C₂H₆) (%)	C _(CO₂) (%)	S _(C₂H₄) (%)	STY _(C₂H₄) (kg h ⁻¹ kg _{cat} ⁻¹)	Utilization (CO ₂) (%)	Ref.
Zn ₃ Cr ₁ /SSZ-13 (oxide)	5	500	5:5:90	7500	9.6	9.5	100	0.045	99.0	This work
Zn ₃ Cr ₁ /SSZ-13 (oxide)	5	550	5:5:90	7500	19.8	18.7	93.0	0.086	94.4	This work
Zn _{9.18} K _{0.74} /nanoSSZ (oxide)	9.18	550	5:5:90	3600	16.7	6.5	94.0	0.035	38.9	5
Zn _{2.92} /NaSSZ-13 (oxide)	2.92	550	5:5:90	3600	23.0	11.5	93.5	0.048	50.0	6
Zn/Na-SSZ-13 (oxide)	8	550	5:5:90	3600	6.0	5.2	100	0.014	86.7	7
5Mo/5CeTiO _x (oxide)	5	550	5:5:90	6000	15.37	-	72.49	0.042	-	8
Mo ₂ C	-	600	10:10:20	24000	2.0	1.0	59.5	0.089	50	9
5Fe/10NiMgZr (oxide)	5	600	8.5:8.5:83	3571	23.0	26.5	69.8	0.061	-	10
5Fe/10NiMgZrO _x (oxide)	5	650	8.6:8.6:82.8	7500	23.3	25.5	90.1	0.169	-	11
5KCr/CeZrO ₂ (oxide)	15	700	20:20:60	6000	32.0	27.3	82.5	0.396	85.3	12
Ni ₁ Fe ₃ -CeO ₂ (Metallic)	1.93	600	10:10:20	24000	3.5	5.9	77.5	0.203	-	13
NiFe-CeO ₂	19.2	600	5:5:10	24000	4.6	6.2	83.0	0.286	-	14

(Metallic)										
Fe- β -Mo ₂ C (Metallic)	1	600	10:10:20	24000	8.7	13.0	62.0	0.405	-	15
Mo ₂ C/SiO ₂	1.42	600	1:1:6	1440	15.5	-	83.5	0.029	-	16
Mo _x C _y /SiO ₂	21.4	650	1:1:2	9400	12.5	-	62.0	0.233	-	17
Fe-Mo _x C _y /SiO ₂	20.5	600	1:1:2	15000	2.3	-	78.0	0.084	-	18
Mo ₂ C/Al ₂ O ₃	20	600	1:1:2	15000	8.3	5.7	62.0	0.241	68.7	19

Response to Reviewer 2

This is a well-written manuscript dealing with an important research topic, the simultaneous upgrading of CO₂ and C₂H₆ to produce CO, C₂H₄ and H₂O. The abundance of C₂H₆ in shale gas reserve makes it an ideal source of H to reduce CO₂; this reaction is potentially advantages over CO₂ reduction using molecular H₂, which generates CO₂ as a byproduct when H₂ is produced from hydrocarbon reforming. Although the reaction of CO₂ + C₂H₆ has been the topic of many recent studies, the current manuscript provides insights into the design of a dual-site Cr-Zn catalyst to achieve nearly 100% C₂H₄ selectivity. The combined experimental and theoretical results also provide useful insights into the reaction mechanism. Before the manuscript can be accepted, the authors should consider the following two suggestions:

Response: We greatly appreciate the reviewer's positive evaluations and constructive suggestions.

1. It is important to compare selectivity at comparable conversions. For example, the results in Table 1 compares the selectivity of Cr/SSZ-13 (5.1% C₂H₆ conversion) and Zn₃Cr1 (19.8% C₂H₆ conversion). Comparing selectivity at conversions that are a factor of 4 different can result in misleading conversions. Because the Arrhenius plot in Figure 1 contains results with less than 10% C₂H₆ conversion, it is likely that the authors already have results at comparable conversions for Cr/SSZ-13 and Zn₃Cr1. The authors should add these results in Table 1 to illustrate that the selectivity is not affected by C₂H₆ conversion.

Response: We agree with the reviewer and are grateful for your valuable suggestions.

We added ethylene selectivity results under comparable conversions for catalyst samples of Cr/SSZ-13 and Zn₃Cr₁/SSZ-13 in Supplementary Table 6. Please check our revision with changes highlighted in red on page S27 in Supplementary Information.

Revision:

Supplementary Table 6. Catalytic performance comparison for Zn₃Cr₁/SSZ-13 and Cr/SSZ-13 at comparable C₂H₆ and CO₂ conversions.

	Zn ₃ Cr ₁ /SSZ-13 (450 °C)	Zn ₃ Cr ₁ /SSZ-13 (475 °C)	Zn ₃ Cr ₁ /SSZ-13 (500 °C)	Cr/SSZ-13 (550 °C)
C ₂ H ₆ Conversion (%)	4.60	6.90	9.60	5.10
CO ₂ conversion (%)	4.50	6.80	9.50	4.90
CO ₂ utilization (%)	98.0	99.0	99.0	96.0
C ₂ H ₄ selectivity (%)	100	100	100	95.4
TOF of C ₂ H ₄ formation (mol mol _{Zn+Cr} ⁻¹ h ⁻¹)	0.19	0.29	0.40	0.20

2. The XANES and EXAFS results provided important information for the authors to support their proposed active sites. However, these measurements appear to be performed under ex-situ conditions. It is possible that the oxidation states of Zn and Cr, as well as the local coordination of Zn, would change under the reaction conditions of CO₂ + C₂H₆. It might be difficult to obtain sufficient beamtime to characterize all the catalysts under in-situ conditions. However, it is important to measure at least one, the best-performing catalyst, under in-situ conditions to confirm that the proposed oxidation states and local coordination environment do not change under reaction conditions.

Response: Thanks for the reviewer’s constructive comments and suggestions.

We agree that it is important to in situ characterize the catalysts to study the oxidation states and local coordination environment of Zn and Cr under reaction conditions. Due to the limited resources of XANES and EXAFS, we are not able to monitor the local coordination environment of our binuclear catalyst, and we used an APXPS to in situ monitor the oxidation states of Zn and Cr under reaction conditions.

As we already stated in Figs. 3b and 3c in the manuscript, “under ultra-high vacuum (UHV)

condition at 550 °C, the ratio of Zn²⁺ (dark blue) and Zn^{δ+} (green) species are 72.4% and 27.6%, respectively; Cr³⁺ (dark red) and Cr⁶⁺ (light blue) were detected with a proportion of 50.6% and 49.4%, respectively. When cofeeding C₂H₆ and CO₂ (ICEC reaction condition), it is observed that the proportion of Zn^{δ+} decreased (from 27.6% to 22.8%) and Cr⁶⁺ ratio increased (from 49.4% to 54.4%).” Above analysis shows the APXPS is capable to monitor the oxidation state changes of our catalyst sample during the reaction.

Then we conducted new control experiment of using APXPS to examine if these changes can be recovered in UHV after pumping out ethane and CO₂. As shown in Supplementary Fig. 29, once that the catalyst was re-exposed to UHV (pumping out ethane and CO₂), the Zn^{δ+} ratio was recovered to 28.0% and Cr⁶⁺ ratio was recovered to 49.0%, which is similar to the initial state of the catalyst.

Therefore, our APXPS results indicate that Zn and Cr in Zn^{δ+}-O-Cr⁶⁺ sites would experience the fast redox cycles of Zn^{δ+} ⇌ Zn²⁺ and Cr⁶⁺ ⇌ Cr³⁺ under the reaction condition, respectively. Binuclear Zn^{δ+}-O-Cr⁶⁺ sites are dynamically stable during ICEC reaction. We hope this explanation clarifies the questions raised by the reviewers. We also added the new APXPS results re-exposed to UHV at 550 °C as Supplementary Fig. 29 and added new analysis.

Revision:

Supplementary Fig. 29. *In situ* ambient pressure X-ray photoelectron spectroscopy (APXPS): **a**, Auger spectra of Zn LMM, and **b**, Cr 2p_{3/2} spectra for Zn₃Cr₁/SSZ-13 upon re-exposure to UHV after pumping out C₂H₆ and CO₂. T = 550 °C.

Supplementary Information, Supplementary Fig. 29, we added the sentence “As shown in supplementary Fig. 29, once that the catalyst was re-exposed to UHV after pumping out C₂H₆ and CO₂, the Zn^{δ+} ratio was recovered to 28.0% and Cr⁶⁺ ratio was recovered to 49.0%, which is similar to the initial state of the catalyst (Fig. 3). This is a further evidence that binuclear Zn^{δ+}-O-Cr⁶⁺ sites are dynamically stable in catalysis”.

Response to Reviewer 3

The submitted manuscript by Yang et al. describes a very interesting piece of work around true catalyst design for a CO₂ activation reaction, which is still not as prominent in the public domain but holds great potential as it does not require the production of green hydrogen as the RWGS does. The authors develop the concept of a bi-nuclear Zn-Cr site, provide a synthesis route, extensive characterization and DFT modelling to elucidate the novel catalyst and its workings. To combine all these aspects in a single paper must be applauded. It is a challenge for a single reviewer to engage with the different areas equally and this reviewer will therefore focus primarily on the testing of the catalyst, interpretation of results and description thereof.

Some comments on the manuscript which are believed to further improve the work.

Response: We appreciate the reviewer “ranks” this work as “a very interesting” study and applauded our paper. That is encouraging.

- 1. The work encompasses on a high level many crucial aspects of modern catalysis research and design. However a thermodynamic analysis is missing. This could be of special interest as the authors operate the reaction with a dilution of 90% inert. This dilution will have an effect of the thermodynamic limit of the reaction.*

Response: Thanks for your comments and suggestions.

We employed Aspen Plus V14 for thermodynamic equilibrium calculations. As shown in Fig. 2c, we added equilibrium conversions of C₂H₆ and CO₂ at different temperatures. To investigate the effect of argon dilution on equilibrium conversions of C₂H₆ and CO₂, we also calculated thermodynamic limit of the reaction in different C₂H₆ and CO₂ concentrations (2.5% CO₂ + 2.5% C₂H₆ + Ar balance; 5% CO₂ + 5% C₂H₆ + Ar balance; 10% CO₂ + 10% C₂H₆ + Ar balance). We found that increasing of CO₂ and C₂H₆ concentrations would decrease the equilibrium conversions. Specifically, at 550 °C, equilibrium conversions of C₂H₆ and CO₂ decrease from 35.0% at C₂H₆

and CO₂ concentrations of 2.5%, 29.4% at C₂H₆ and CO₂ concentrations of 5%, to 24.5% at C₂H₆ and CO₂ concentrations of 10%. We also updated these results in Supplementary Information (Supplementary Fig. 20).

Revisions:

Fig. 2c. Reaction temperature dependent performance of Zn₃Cr₁/SSZ-13 catalyst.

Supplementary Fig. 20. Calculated equilibrium conversions of C₂H₆ and CO₂ for iso-stoichiometric co-conversion of ethane and CO₂ (ICEC) in different concentrations (2.5%, 5%, and 10%).

2. *It is also challenging for the reader to get a clear picture of the performance of the produced catalyst as the authors continuously switch between experimental performance at 500 and 550C. Stability information is provided to some extent at the lower temperature, but not at 550C. It is also not clear in most cases at what time on stream performance data is collected. It is well reported that CO₂-ODH catalysts commonly suffer from deactivation over the initial 12 to 24 hours, so catalytic performance in the initial hours TOS might not be sufficiently representative.*

Response: Thanks for the reviewer's comments.

We compared the performance of all catalyst samples under 550 °C and conducted the stability testing under 500 °C (Fig. 2 and Supplementary Figs.19-23). Under 500 °C, the Zn₃Cr₁/SSZ-13 catalyst exhibited the highest CO₂ utilization and C₂H₄ selectivity with decent CO₂ (9.5%) and C₂H₆ conversions (9.6 %). The CO₂ utilization and C₂H₄ selectivity are the key factors for us to demonstrate the concept of iso-stoichiometric co-conversion of ethane and CO₂ (ICEC), therefore, we chose 500 °C for the stability testing. And a relatively low reaction temperature could reduce the influences of metal oxide cluster sintering on these key factors and catalyst stability.

The performance of all catalyst samples in Fig. 2a were collected in the initial states (within 1h). Comparing the performance of a group of catalyst samples under steady-state operation is challenging given the transient behavior of the catalyst and given that the different catalyst has various induction stage. Furthermore, the catalytic performances were collected in the initial states in this study makes the performance comparison with previous studies meaningful. [Appl. Catal. B Environ. 2022, 304, 120947, ACS Catal. 2021, 11, 2819–2830].

As shown in Supplementary Fig. 24, we also observed a slight deactivation in the induction stage over the initial 15 h. After that, the catalyst exhibited almost stable C₂H₆ and CO₂ conversions (which slightly decreased from 7.5% in 15h to 7.0% in 50h). We added the above analysis to Supplementary Methods (on Page S2).

Revision:

Supplementary Methods of Supplementary Information, in Performance tests, we added the

sentence “The performance of all catalyst samples in Fig. 2a and Supplementary Fig. 19 were collected in the initial states under 550 °C. The stability test was conducted under reaction temperature of 500 °C, with the highest U_{CO_2} and C_2H_4 selectivity at the decent reactant conversions, over the $Zn_3Cr_1/SSZ-13$ catalyst”.

3. *We would also urge the authors to provide more detail on the product distribution. What is the balance in selectivity if 100% is not reached. I would also prefer if the exact value of the CO₂ utilization is provided and not the approximately 100%. The authors provide sufficient accuracy in the other data presented.*

Response: Thanks for your comments and suggestions. According to our product analysis results (in the table below), the by-product is methane (CH_4) and no other compounds such as acetylene or aromatics were detected. We have added the description of the product distribution to Supplementary Fig. 19. Besides, we provided exact value of 99.0 % for CO_2 utilization (the term of “ CO_2 utilization” is revised to “utilization of converted CO_2 (U_{CO_2})” according to reviewer 1’s suggestion) to replace ~100% and updated this result throughout the manuscript.

Table X. Product distribution for ICEC reaction on our catalyst samples.

Catalysts	T (°C)	C_2H_6 conversion (%)	CO_2 conversion (%)	Utilization (CO_2) (%)	C_2H_4 selectivity (%)	CH_4 selectivity (%)	C_2H_2 Selectivity (%)	C_{3+} selectivity (%)
$Zn_3Cr_1/SSZ-13$	500	9.6	9.5	99.0	100	0	0	0
$Zn_3Cr_1/SSZ-13$	550	19.8	18.7	94.4	93.0	7.0	0	0
$Cr/SSZ-13$	550	5.1	4.9	96.0	95.4	4.6	0	0
$Zn_1Cr_2/SSZ-13$	550	9.5	9.1	96.0	96.0	4.0	0	0
$Zn_1Cr_1/SSZ-13$	550	13.3	12.2	92.0	94.0	6.0	0	0
$Zn_2Cr_1/SSZ-13$	550	14.7	13.6	92.5	95.0	5.0	0	0
$Zn_4Cr_1/SSZ-13$	550	14.4	11.8	81.5	94.0	6.0	0	0
$Zn/SSZ-13$	550	12.2	10.2	83.6	93.0	7.0	0	0

Revision:

Supplementary Information, in Supplementary Fig. 19, we added the sentence “According to product analysis results, the by-product is methane (CH_4) when C_2H_4 selectivity is less than 100%”.

4. In Table S4, the authors compare performances of their own design with other reported catalyst systems. One class that has recently attracted some attention, namely molybdenum carbides is only listed in its bulk form. There have been a number of publications recently from different groups suggesting that supported molybdenum carbide does perform much better. It would be beneficial for the manuscript and future readers if this development is reflected.

Response: Thanks for your comments and suggestions.

We updated the Table S4 (now refer as Supplementary Table 7) by adding more previous studies including the supported molybdenum carbide catalysts for ICEC reaction. Please check the revisions highlighted in Supplementary Table 7 below. Compared with the previous studies, our non-noble-metal and atomically synergistic catalyst displayed advantages in achieving higher ethylene selectivity and U_{CO_2} in ICEC.

Revision:

Supplementary Table 7. Summarization of catalytic behaviors over previously reported catalysts at CO_2/C_2H_6 ratio of 1.

Catalysts	Loading (wt%)	T (°C)	Reactants compositions ($C_2H_6 : CO_2 : Ar/He$)	WHSV ($mL g_{cat}^{-1} h^{-1}$)	$C_{(C_2H_6)}$ (%)	$C_{(CO_2)}$ (%)	$S_{(C_2H_4)}$ (%)	STY _(C₂H₄) ($kg h^{-1} kg_{cat}^{-1}$)	Utilization (CO_2) (%)	Ref.
$Zn_3Cr_1/SSZ-13$ (oxide)	5	500	5:5:90	7500	9.6	9.5	100	0.045	99.0	This work
$Zn_3Cr_1/SSZ-13$ (oxide)	5	550	5:5:90	7500	19.8	18.7	93.0	0.086	94.4	This work
$Zn_{9.18}K_{0.74}/nanoSSZ$ (oxide)	9.18	550	5:5:90	3600	16.7	6.5	94.0	0.035	38.9	5
$Zn_{2.92}/NaSSZ-13$ (oxide)	2.92	550	5:5:90	3600	23.0	11.5	93.5	0.048	50.0	6
$Zn/Na-SSZ-13$ (oxide)	8	550	5:5:90	3600	6.0	5.2	100	0.014	86.7	7
$5Mo/5CeTiO_x$ (oxide)	5	550	5:5:90	6000	15.37	-	72.49	0.042	-	8
Mo_2C	-	600	10:10:20	24000	2.0	1.0	59.5	0.089	50	9

5Fe/10NiMgZr (oxide)	5	600	8.5:8.5:83	3571	23.0	26.5	69.8	0.061	-	10
5Fe/10NiMgZrOx (oxide)	5	650	8.6:8.6:82.8	7500	23.3	25.5	90.1	0.169	-	11
5KCr/CeZrO ₂ (oxide)	15	700	20:20:60	6000	32.0	27.3	82.5	0.396	85.3	12
Ni ₁ Fe ₃ -CeO ₂ (Metallic)	1.93	600	10:10:20	24000	3.5	5.9	77.5	0.203	-	13
NiFe-CeO ₂ (Metallic)	19.2	600	5:5:10	24000	4.6	6.2	83.0	0.286	-	14
Fe-β-Mo ₂ C (Metallic)	1	600	10:10:20	24000	8.7	13.0	62.0	0.405	-	15
Mo ₂ C/SiO ₂	1.42	600	1:1:6	1440	15.5	-	83.5	0.029	-	16
Mo _x C _y /SiO ₂	21.4	650	1:1:2	9400	12.5	-	62.0	0.233	-	17
Fe-Mo _x C _y /SiO ₂	20.5	600	1:1:2	15000	2.3	-	78.0	0.084	-	18
Mo ₂ C/Al ₂ O ₃	20	600	1:1:2	15000	8.3	5.7	62.0	0.241	68.7	19

Response to Reviewer 4

The paper focuses on atomically synergistic binuclear-site Zn-Cr zeolitic catalyst for co-conversion of ethane and CO₂. The study presents advanced material characterization data, modelling, catalytic performances in co-conversion of ethane and CO₂. The insight in material, the discussions involving modelling and advanced characterization, the selection of the reaction and the reaction conditions to highlight the synergy between Cr-Zn, and the observed phenomena are interesting and warrant publication. However, there are several comments raised that require attention.

Response: We greatly appreciate the reviewer “ranks” our work as “warrant publication”. We also thank the reviewer for the insightful and constructive comments.

1. *The authors report synthetic procedure, XRD and a lot of high-level characterization data of the metal-phases by XAFS, XPS, HRTEM and HAADF-STEM etc. This information is only*

valuable if the basic characterization of the materials would be given. However, the article is lacking the information about basic characteristics which is typically used by experts in the field, especially for the zeolitic phase. So, the readers of the article will not be able to have any points of comparison of the classical prior art materials with the one, which is reported in this contribution. Without this information, this contribution would not be very useful for the scientific community (difficult to reproduce the data).

One would expect to see values for

- basic porosity characteristics of samples with different metal ratios,*
- form of zeolite before modification (H^+ , NH_4^+ , Ca, K etc),*
- crystal size of zeolite,*
- acidity (any characteristics, i.e. OH-groups by FTIR or NMR).*

These values also are fundamental for the explanation of the catalytic data.

Response: Your constructive feedback and suggestions are greatly appreciated, and we are in complete agreement with your points.

We performed N_2 physisorption, TEM, and attenuated total reflection Fourier transform infrared (ATR-FTIR) measurements to study the physical and chemical properties of the zeolite SSZ-13 and catalyst samples.

In this work, we used commercial H^+ -form SSZ-13 (SiO_2/Al_2O_3 molar ratio of 30; ACS Materials Inc.) as parent material for catalyst fabrication. Textural properties of parent SSZ-13 zeolite and synthesized catalysts determined by N_2 physisorption are shown in the Table below. The surface area and pore volume of the parent SSZ-13 zeolite and catalyst samples are comparable, which indicates that the deposition of Zn and Cr species has little influence on pore structure of zeolites. We updated the results as Supplementary Table 1. Furthermore, we updated TEM results in Supplementary Fig. 1. It suggests that $(H^+)SSZ-13$ shows cubic morphology with an average crystallite size in the $\sim 1-2 \mu m$ range. ATR-FTIR spectra of the samples are shown in Supplementary Fig. 2. The bands at 3744, 3655, and $\sim 3475 \text{ cm}^{-1}$ correspond to isolated hydroxyl group (OH), bridging OH, and OH nests, respectively [Appl. Catal. B Environ. 2022, 304, 120947; Nature, 2021, 599, 234-238]. After Na^+ modification on $(H^+)SSZ-13$, the peak intensity of OH nests significantly decreased, indicating the weakened Brønsted acidity. Moreover, the deposition of ZnCrOx species on $Zn_3Cr_1/SSZ-13$ further reduces the peak intensity of bridging OH and OH nests, indicating a further reduced surface Brønsted acidity.

Revision:

Supplementary Table 1. Physical characteristics of parent zeolite and as-synthesized catalyst samples.

Catalyst ^a	S _{BET} ^b (m ² g ⁻¹)	S _{micro} ^c (m ² g ⁻¹)	V _{pore} (cm ³ g ⁻¹)
(H ⁺) SSZ-13	488	463	0.253
(Na ⁺) SSZ-13	445	432	0.223
Cr/SSZ-13	504	482	0.267
Zn ₁ Cr ₂ /SSZ-13	474	451	0.247
Zn ₁ Cr ₁ /SSZ-13	486	459	0.256
Zn ₂ Cr ₁ /SSZ-13	497	475	0.260
Zn ₃ Cr ₁ /SSZ-13	499	472	0.263
Zn ₄ Cr ₁ /SSZ-13	496	473	0.260
Zn/SSZ-13	498	477	0.260

^aSSZ-13 support with SiO₂/Al₂O₃ molar ratio of 30 purchased from ACS Material Inc. ^bBrunauer-Emmett-Teller (BET) method applied to the N₂ isotherm. ^ct-plot method applied to the N₂ isotherm

Supplementary Information, Supplementary Table 1, we added the sentence “The surface area and pore volume of the parent SSZ-13 zeolite and catalyst samples are comparable, which indicates that the deposition of Zn and Cr species has little influence on pore structure of zeolites”.

Supplementary Fig. 1. TEM images of (H⁺)SSZ-13 zeolite support.

Supplementary Information, Supplementary Fig. 1, we added the sentence “TEM results suggest that (H⁺)SSZ-13 shows cubic morphology with an average crystallite size in the ~1-2 μm range”.

Supplementary Fig. 2. ATR-FTIR spectra of (H⁺)SSZ-13, (Na⁺)SSZ-13 and as-synthesized Zn₃Cr₁/SSZ-13 catalyst.

Supplementary Information, Supplementary Fig. 2, we added the sentence “The bands at 3744, 3655, and ~3475 cm⁻¹ correspond to isolated hydroxyl group (OH), bridging OH, and OH nests, respectively^{1,2}. After Na⁺ modification on (H⁺)SSZ-13, the peak intensity of OH nests significantly decreased, indicating the weakened Brønsted acidity. Moreover, the deposition of ZnCrOx species on Zn₃Cr₁/SSZ-13 further reduces the peak intensity of bridging OH and OH nests, indicating a further reduced surface Brønsted acidity”.

- The concentration of the metals deposited on the samples was not measured (not controlled) by elemental analysis. It was assumed that the total amount of Zn and Cr, which was added to the synthetic mixture as acetates, would be deposited on the zeolites. It is surprising to perform so many advanced measurements for the samples without controlling / knowing the exact chemical composition of the materials.*

Response: Thanks for your comments and suggestions.

We performed X-ray fluorescence (XRF) and X-ray photoelectron spectroscopy (XPS) for

elemental analysis of the bulk sample and catalyst surface, respectively. XRF and XPS results show that surface Zn/Cr ratios is basically consistent with bulk Zn/Cr stoichiometry for Zn_xCr_y catalyst samples ($x/y= 4/1, 3/1, \text{ and } 1$). We updated these results as Supplementary Table 2.

Revision:

Supplementary Table 2. Bulk and surface chemical compositions of Zn and Cr in as-synthesized catalysts determined by X-ray fluorescence (XRF) and XPS, respectively.

Sample	Atomic ratio ^a			
	Zn	Cr	Al	Si
SSZ-13	-	-	3.5	96.5
Zn/SSZ-13	8.7	-	3.7	87.7
Zn_4Cr_1 /SSZ-13	7.5	1.8	3.8	86.9
Zn_3Cr_1 /SSZ-13	6.6	2.0	4.1	87.3
Zn_1Cr_1 /SSZ-13	4.9	3.7	3.8	87.6
Cr/SSZ-13	-	8.6	3.5	87.9

^aDetermined by XRF

	Zn_1Cr_1 /SSZ-13	Zn_3Cr_1 /SSZ-13	Zn_4Cr_1 /SSZ-13
Surface Zn/Cr atomic ratio ^a	1.2	3.2	3.8

^aDetermined by XPS

Supplementary Information, Supplementary Table 2, we added the sentence “XRF and XPS results show that surface Zn/Cr ratios is basically consistent with bulk Zn/Cr stoichiometry for Zn_xCr_y catalyst samples ($x/y= 4/1, 3/1, \text{ and } 1$)”.

3. The authors report significantly better catalytic activity, selectivity, stability in co-conversion of ethane and CO₂ vs. the recent publications for the same reaction under similar operating conditions (*Appl. Catal. B Environ.* 304, 120947 (2022), *ACS Catal.* 11, 2819–2830). However, the catalytic data were produced at very low partial pressures of ethane and CO₂. The reported mass balance is 100% ±2 % (based on internal standard Ar). However, the internal standard was used in 16 times excess relative to the ethane and CO₂ in the feed (5/5/80). Under such a

condition, the variation of flow of the internal standard of 1% would lead to a significant change in measured catalytic performances. So, under such extreme conditions, a comparison of the data between different labs should be taken with precautions.

Nevertheless, under the reported conditions, the obtained results look not in a conflict with thermodynamics equilibrium (ethylene yield). One should also mention that the catalytic performance is reported at 500-550oC, which is 100oC-250oC lower than the typical temperature range in the classical literature for the same reaction (Energy & Fuels 2004, 18, 1126-1139, table 2). The reason for that is in the "extreme" catalytic conditions (very low partial pressure). These conditions may be very beneficial for demonstrating the synergistic effect between Zn and Cr but wasn't used in the classic literature because of the limited practical significance. So, despite the catalysts show a very good performances, this is a not yet a breakthrough in CO₂ utilization.

One should see the reaction as fine characterization of materials properties and as a demonstration of a synergy between Zn-Cr on SSZ-13 zeolite. So, the material part remains interesting if the additional basic characterization would be provided.

Response: We deeply appreciate your positive and insightful comments.

In order to prevent measurement errors in catalyst performance testing stemming from possible fluctuations in the carrier gas Ar flow rate, we performed a control experiment with C₂H₆/CO₂/Ar/He ratio of 1:1:2:16. Furthermore, we also tested the ICEC reaction under C₂H₆/CO₂/Ar ratio of 1:1:2 (molar ratio) with high partial pressure of C₂H₆ and CO₂. Both catalytic performance evaluations were conducted under a reaction temperature of 500 °C. We updated these results with our previous result (with C₂H₆/CO₂/Ar ratio of 1:1:18, under 500 °C) in the Table below. When comparing the results in first two columns, no obvious difference was detected, especially for the ethylene selectivity and CO₂ utilization. By comparing the third column with the first one, when we only decreased the flow rate of Ar to achieve a C₂H₆/CO₂/Ar ratio of 1:1:2, the C₂H₆ conversion and CO₂ conversion were slightly increased due to the lower WHSV. The U_{CO₂} (98.0%) and C₂H₄ selectivity (96.2%) is still much higher than previous studies (Supplementary Table 7) even if they were slightly decreased. We hope our analysis can address the reviewer's concerns.

We updated the thermodynamic analysis results in Fig. 2c, which show that C₂H₆ and CO₂ conversions of our study are within thermodynamics equilibriums.

In this study, we demonstrated a concept of ICEC reaction based on atomically synergistic catalyst design and fabrication. The studies presented in [Energy & Fuels 2004, 18, 1126-1139] primarily focus on the dehydrogenation of ethane to produce highly valuable ethylene, with relatively limited attention given to the aspects of CO₂/Ethane feeding ratio and CO₂ utilization. In Supplementary Table 7 and the Table below, we demonstrated that the Zn₃Cr₁/SSZ-13 catalyst displayed the highest U_{CO₂} and C₂H₄ selectivity compared with previous studies under the ICEC reaction conditions.

Table X. ICEC performance over Zn₃Cr₁/SSZ-13 at different catalytic conditions.

	C ₂ H ₆ :CO ₂ :Ar = 1:1:18 WHSV= 7500 mL g _{cat} ⁻¹ h ⁻¹	C ₂ H ₆ :CO ₂ :Ar:He = 1:1:2:16 WHSV= 7500 mL g _{cat} ⁻¹ h ⁻¹	C ₂ H ₆ :CO ₂ :Ar = 1:1:2 WHSV=1500 mL g _{cat} ⁻¹ h ⁻¹
C ₂ H ₆ conversion	9.6%	9.5%	9.9%
CO ₂ conversion	9.5%	9.4%	9.7%
U _{CO₂}	99.0%	99.0%	98.0%
C ₂ H ₄ selectivity	100%	100%	96.2%

Revision:

Fig. 2c. Reaction temperature dependent performance of Zn₃Cr₁/SSZ-13 catalyst.

REVIEWERS' COMMENTS

Reviewer #1 (Remarks to the Author):

The authors have very carefully addressed all my comments. I am very satisfied from their effort and the manuscript deserves to be published in Nature Communications journal.

Some minor comments to be addressed before publication:

- 1) In methods section, please include the units for the τ used for STY calculation
- 2) Table 7 from the Supplementary: what are the units for the reactants composition column? Is it ratio or ml/min? The authors should double-check the reference 13 and 14, since the 10:10:20 refer to flowrate (ml/min). Figure 1 from reference 13 shows the performance for 100 mg catalyst. Can the authors double-check the reported STY? It looks very high considering the C_2H_6 conversion and C_2H_4 selectivity.

Reviewer #2 (Remarks to the Author):

All my comments have been adequately addressed and clarified. I recommend acceptance.

Reviewer #3 (Remarks to the Author):

The authors have extensively responded to the comments and suggestions raised to the initial manuscript submission. I feel the work has significantly benefited from the process and is now suitable for publication.

Point-by-point response to the reviewers' comments:

Reviewer #1:

The authors have very carefully addressed all my comments. I am very satisfied from their effort and the manuscript deserves to be published in Nature Communications journal.

Response: We greatly appreciate the reviewer's positive evaluations.

Some minor comments to be addressed before publication:

1) In methods section, please include the units for the m_{cat} used for STY calculation.

Response: Thanks for the comment. The unit for the m_{cat} used for STY calculation is kilogram (kg), which is already added in the section of Methods.

2) Table 7 from the Supplementary: what are the units for the reactants composition column? Is it ratio or ml/min? The authors should double-check the reference 13 and 14, since the 10:10:20 refer to flowrate (ml/min). Figure 1 from reference 13 shows the performance for 100 mg catalyst. Can the authors double-check the reported STY? It looks very high considering the C₂H₆ conversion and C₂H₄ selectivity.

Response: Sorry for the confusion.

The units for the reactants composition column is molar ratio. We have unified the units of reactants composition column in Supplementary Table 7 and updated the supplementary note.

To compare the catalytic performance of this work with previous studies, a space time yield (STY) of C₂H₄ formation in Supplementary Table 7 were calculated as follows (where $n_{\text{C}_2\text{H}_4 \text{ output}}$ denotes molar flow of generated C₂H₄ (mol/min), $M_{\text{C}_2\text{H}_4}$ is the molecular weight of C₂H₄ (28 g/mol) and m_{cat} stands for catalyst mass (kg)):

$$\text{STY} = \frac{n_{\text{C}_2\text{H}_4 \text{ output}} \times M_{\text{C}_2\text{H}_4} \times 60}{1000 \times m_{\text{cat}}}$$

Following the reviewer's suggestion, we double checked the STY of reference 13 with Ni₁Fe₃/CeO₂ sample:

$$\begin{aligned} \text{STY} &= \frac{(40 \text{ mL/min} \times 25\%/1000 \text{ (mL/L)}/22.4 \text{ (L/mol)} \times 3.5\% \times 77.5\%) \times 28 \text{ g/mol} \times 60 \text{ min/h}}{1000 \text{ g/kg} \times 0.0001 \text{ kg}} \\ &= \frac{0.000012109 \text{ mol/min} \times 28 \text{ g/mol} \times 60 \text{ min/h}}{1000 \text{ g/kg} \times 0.0001 \text{ kg}} \\ &= \frac{0.020343 \text{ g/h}}{1000 \text{ g/kg} \times 0.0001 \text{ kg}} \\ &= 0.20343 \text{ kg h}^{-1} \text{ kg}_{\text{cat}}^{-1} \end{aligned}$$

We also attached calculation results and methods of the STY of C₂H₄ formation (STY_{C₂H₄}) in ref. 13 below. From their summarization results (Table S2), the STY_{C₂H₄} on Ni₁Fe₃/CeO₂ is 101.7 μmol min⁻¹ g_{cat}⁻¹ (equal to 0.171 kg h⁻¹ kg_{cat}⁻¹), which is slightly lower than the results from our calculation methods (0.203 kg h⁻¹ kg_{cat}⁻¹).

⁻¹_{cat}). From equation S8 below (copied from SI of ref. 13), we found that the calculation of STY_{C₂H₄} in ref. 13 is based on detected flow rate of C₂H₄ generation. Considering that there is an increase in the volume when ethane dry reforming reaction (C₂H₆ + 2CO₂ → 4CO + 3H₂) takes place, it is possible that the GC-detected C₂H₄ concentration is affected by the increased volume. We think this possibly account for the STY_{C₂H₄} difference in two methods.

Table S2. Summary of flow reactor results for CO₂ + C₂H₆ reactions (1:1 ratio, 10 mL/min C₂H₆ + 10 mL/min CO₂ + 20 mL/min Ar, ~100 mg catalyst) at 873 K. Values of conversion, selectivity, yield, TOF and STY were calculated by averaging data points between 11-13 h on stream.

Catalyst	Conversion, %		Selectivity, %		Yield, %		TOF, time/site/min		STY, μmol/g _{catalyst} /min			
	CO ₂	C ₂ H ₆	C ₂ H ₄	CO	C ₂ H ₄	CO	CO ₂	C ₂ H ₆	C ₂ H ₄	CO	H ₂	CH ₄
Fe ₃ /CeO ₂	1.7	0.5	38.3	61.1	0.2	0.3	1.2	0.37	7.7	94.1	3.8	0.3
Ni _{0.5} Fe ₃ /CeO ₂	5.4	2.8	66.7	33.1	1.8	0.9	3.4	1.8	66.6	276.2	24.4	0.6
Ni₁Fe₃/CeO₂	5.9	3.5	77.5	22.3	2.7	0.8	4.0	2.3	101.7	299.7	27.5	0.7
Ni ₂ Fe ₃ /CeO ₂	7.4	4.3	72.6	27.1	3.1	1.2	4.5	2.6	113.1	377.4	46.6	0.9
Ni ₃ Fe ₃ /CeO ₂	10.4	4.4	46.9	52.8	2.0	2.3	6.6	2.8	76.4	595.3	89.0	1.0
Ni ₃ Fe ₂ /CeO ₂	22.8	7.5	3.2	96.6	0.2	7.2	20.8	6.8	9.1	1481.9	448.9	1.0
Ni ₃ Fe ₁ /CeO ₂	41.3	16.7	0.70	98.6	0.1	16.5	59.9	24.2	4.5	2960.7	1371.1	9.6
Ni ₃ Fe _{0.5} /CeO ₂	38.4	15.4	0.79	98.8	0.1	15.2	74.7	29.9	4.7	2764.7	1281.8	4.7
Ni ₃ /CeO ₂	35.3	14.0	0.89	98.8	0.1	13.8	93.7	37.2	4.7	2475.5	1095.5	3.4

Calculation equations in Ref. 13:

$$X_{\text{reactant}} = \frac{F_{\text{reactant}}^{\text{inlet}} - F_{\text{reactant}}^{\text{outlet}}}{F_{\text{reactant}}^{\text{inlet}}} \quad (\text{S1})$$

$$Y_{\text{C}_2\text{H}_4} = \frac{F_{\text{C}_2\text{H}_4}^{\text{outlet}}}{F_{\text{C}_2\text{H}_6}^{\text{inlet}}} \quad (\text{S2})$$

$$Y_{\text{CO}} = \frac{F_{\text{CO}}^{\text{outlet}}}{2 \times F_{\text{C}_2\text{H}_6}^{\text{inlet}}} \quad (\text{S3})$$

Here, the amount of CO produced for C₂H₆ can be calculated based on the oxygen balance,

$$F_{\text{CO}}^{\text{outlet}} = F_{\text{CO}}^{\text{outlet}} - \left(\frac{F_{\text{CO}}^{\text{outlet}} + F_{\text{H}_2\text{O}}^{\text{outlet}}}{2} \right) \quad (\text{S4})$$

$$S_{\text{C}_2\text{H}_4} = \frac{Y_{\text{C}_2\text{H}_4}}{X_{\text{C}_2\text{H}_6}} \quad (\text{S5})$$

$$S_{\text{CO}} = \frac{Y_{\text{CO}}}{X_{\text{C}_2\text{H}_6}} \quad (\text{S6})$$

$$\text{TOF} = \frac{F_{\text{reactant}}^{\text{inlet}} \times X}{U_{\text{CO}} \times W_{\text{catalyst}}}, \text{ time/site/min} \quad (\text{S7})$$

$$\text{STY}_{\text{product}} = \frac{F_{\text{product}}^{\text{outlet}}}{W_{\text{catalyst}}}, \text{ } \mu\text{mol/g}_{\text{catalyst}}/\text{min} \quad (\text{S8})$$

where, F is the flow rate of reactant, mol/min; U_{CO} is the CO uptake value, μmol CO/g; W is the weight of catalyst used, g.

Reviewer #2:

All my comments have been adequately addressed and clarified. I recommend acceptance.

Reviewer #3:

The authors have extensively responded to the comments and suggestions raised to the initial manuscript submission. I feel the work has significantly benefited from the process and is now suitable for publication.

Response: We greatly appreciate the reviewers' (#2 and #3) positive evaluations.